# Extensive and diverse patterns of cell death sculpt neural networks in insects

**Sinziana Pop[1], Chin-Lin Chen[2], Connor J Sproston[1], Shu Kondo[3], Pavan Ramdya[2]\*, Darren W Williams[1]\***

[1]Centre for Developmental Neurobiology, King's College London, London, United Kingdom; [2]Neuroengineering Laboratory, Brain Mind Institute and Institute of Bioengineering, École Polytechnique Fédérale de Lausanne, Lausanne, Switzerland; [3]Genetic Strains Research Center, National Institute of Genetics, Shizuoka, Japan

**Abstract** Changes to the structure and function of neural networks are thought to underlie the evolutionary adaptation of animal behaviours. Among the many developmental phenomena that generate change programmed cell death (PCD) appears to play a key role. We show that cell death occurs continuously throughout insect neurogenesis and happens soon after neurons are born. Mimicking an evolutionary role for increasing cell numbers, we artificially block PCD in the medial neuroblast lineage in *Drosophila melanogaster*, which results in the production of 'undead' neurons with complex arborisations and distinct neurotransmitter identities. Activation of these 'undead' neurons and recordings of neural activity in behaving animals demonstrate that they are functional. Focusing on two dipterans which have lost flight during evolution we reveal that reductions in populations of flight interneurons are likely caused by increased cell death during development. Our findings suggest that the evolutionary modulation of death-based patterning could generate novel network configurations.

**\*For correspondence:**
pavan.ramdya@epfl.ch (PR);
darren.williams@kcl.ac.uk (DWW)

**Competing interests:** The authors declare that no competing interests exist.

## Introduction

Nervous systems are exquisitely adapted to the biomechanical and ecological environments in which they operate. How they evolve to be this way is largely unknown. Such changes can occur through modifications in receptor tuning, transmitter/receptor repertoires, neuronal excitability, neuromodulation, structural connectivity, or in the number of neurons within specific regions of the central nervous system (CNS). The differences seen in networks, over an evolutionary timescale, ultimately result from heritable changes in developmental processes (*Horder, 1989*). Advancing our knowledge of the mechanisms of neural development using comparative approaches will help us understand how specific elements can be modified, how new 'circuits' and behaviours evolve, and will ultimately lead to a better understanding of how nervous systems function (*Ramdya and Benton, 2010*). Studies comparing the nervous systems of mammalian species that occupy diverse ecological niches reveal clear differences in the number of cells within homologous brain regions (*Herculano-Houzel et al., 2014*). Such differences have occurred either through expansion or reduction of specific cell populations, through changes in proliferation or apoptotic programmed cell death (PCD) during development (*Charvet et al., 2011*). Most studies of nervous system evolution have focused on stem cell identity and the role of differential proliferation dynamics (*Biffar and Stollewerk, 2014*; *Rakic, 2009*; *Truman and Ball, 1998*). While one recent study has elegantly shown a role for PCD in the evolution of peripheral olfactory sensory neurons in drosophilids and mosquitoes (*Prieto-Godino et al., 2020*), how changes in cell death can modify central circuits still remains an open question.

In insects, the number and arrangement of neural progenitor cells that generate central neurons (termed neuroblasts, NBs) are highly conserved despite the remarkable diversity of insect body plans

**eLife digest** Just like a sculptor chips away at a block of granite to make a statue, the nervous system reaches its mature state by eliminating neurons during development through a process known as programmed cell death.

In vertebrates, this mechanism often involves newly born neurons shrivelling away and dying if they fail to connect with others during development. Most studies in insects have focused on the death of neurons that occurs at metamorphosis, during the transition between larva to adult, when cells which are no longer needed in the new life stage are eliminated.

Pop et al. harnessed a newly designed genetic probe to point out that, in fruit flies, programmed cell death of neurons at metamorphosis is not the main mechanism through which cells die. Rather, the majority of cell death takes place as soon as neurons are born throughout all larval stages, when most of the adult nervous system is built. To gain further insight into the role of this 'early' cell death, the neurons were stopped from dying, showing that these cells were able to reach maturity and function. Together, these results suggest that early cell death may be a mechanism fine-tuned by evolution to shape the many and varied nervous systems of insects.

To explore this, Pop et al. looked for hints of early cell death in relatives of fruit flies that are unable to fly: the swift lousefly and the bee lousefly. This analysis showed that early cell death is likely to occur in these two insects, but it follows different patterns than in the fruit fly, potentially targeting the neurons that would have controlled flight in these flies' ancestors.

Brains are the product of evolution: learning how neurons change their connections and adapt could help us understand how the brain works in health and disease. This knowledge may also be relevant to work on artificial intelligence, a discipline that often bases the building blocks and connections in artificial 'brains' on how neurons communicate with one another.

and behaviours (*Bate, 1976*; *Biffar and Stollewerk, 2014*; *Booker and Truman, 1987*; *Doe, 1992*; *Doe and Goodman, 1985*; *Hartenstein and Campos-Ortega, 1984*; *Nordlander and Edwards, 1969*; *Shepherd and Bate, 1990*; *Tamarelle et al., 1985*; *Truman, 1996*; *Truman and Ball, 1998*; *Truman and Bate, 1988*; *Wheeler, 1891*). In the ventral nerve cord (VNC – functionally equivalent to the vertebrate spinal cord) all but one NBs are arranged in a bilaterally symmetric array across the midline, while an unpaired, single medial neuroblast (MNB) stands out in the posterior end of each segment (*Figure 1A,B*).

In the *Drosophila* embryo a first wave of neurogenesis generates the larval nervous system after which the majority of NBs become quiescent. Following reactivation from quiescence NBs produce neurons throughout larval life until the early pupal stages (*Booker and Truman, 1987*; *Truman and Bate, 1988*). These postembryonic neurons – which make up most of the adult CNS – extend simple neuritic processes into the neuropil and stall until the pupal-adult transition when they grow complex arborisations, synapsing with their target cells (*Truman, 1990*). In the VNC, NBs bud off a ganglion mother cell (GMC) which undergoes a terminal division to generate two neurons with distinctly different cell fates (an A cell and a B cell). As the A and B cells result from a single division, one cannot be produced without the other. After several rounds of GMC divisions, a lineage produced by a single NB is composed of two half-lineages: 'hemilineage A' made up of all the A cells and 'hemilineage B' made up of B cells (*Figure 1C*). Hemilineages act as functional units in adult flies (*Harris et al., 2015*; *Lacin et al., 2019*; *Lin et al., 2010*; *Shepherd et al., 2016*; *Shepherd et al., 2019*; *Truman et al., 2010*; *Truman et al., 2004*). For example, in the MNB lineage, hemilineage A cells mature into GABAergic local interneurons while hemilineage B cells become efferent octopaminergic neurons. Our previous work showed that a common fate of postembryonic neurons is PCD affecting approximately 40% of VNC hemilineages (*Figure 1D,E*; *Truman et al., 2010*), this is also seen in the brain (*Bertet et al., 2014*; *Kumar et al., 2009*; *Lin et al., 2010*). The pattern of PCD is stereotypical and targets the same hemilineages across individuals. Taken together, the breadth of PCD suggests it plays a major role in shaping the final makeup of the adult nervous system, while its stereotypy points towards a heritable genetic basis. We therefore propose that changes in neural circuits may result from heritable alterations in the extent and pattern of PCD in hemilineages.

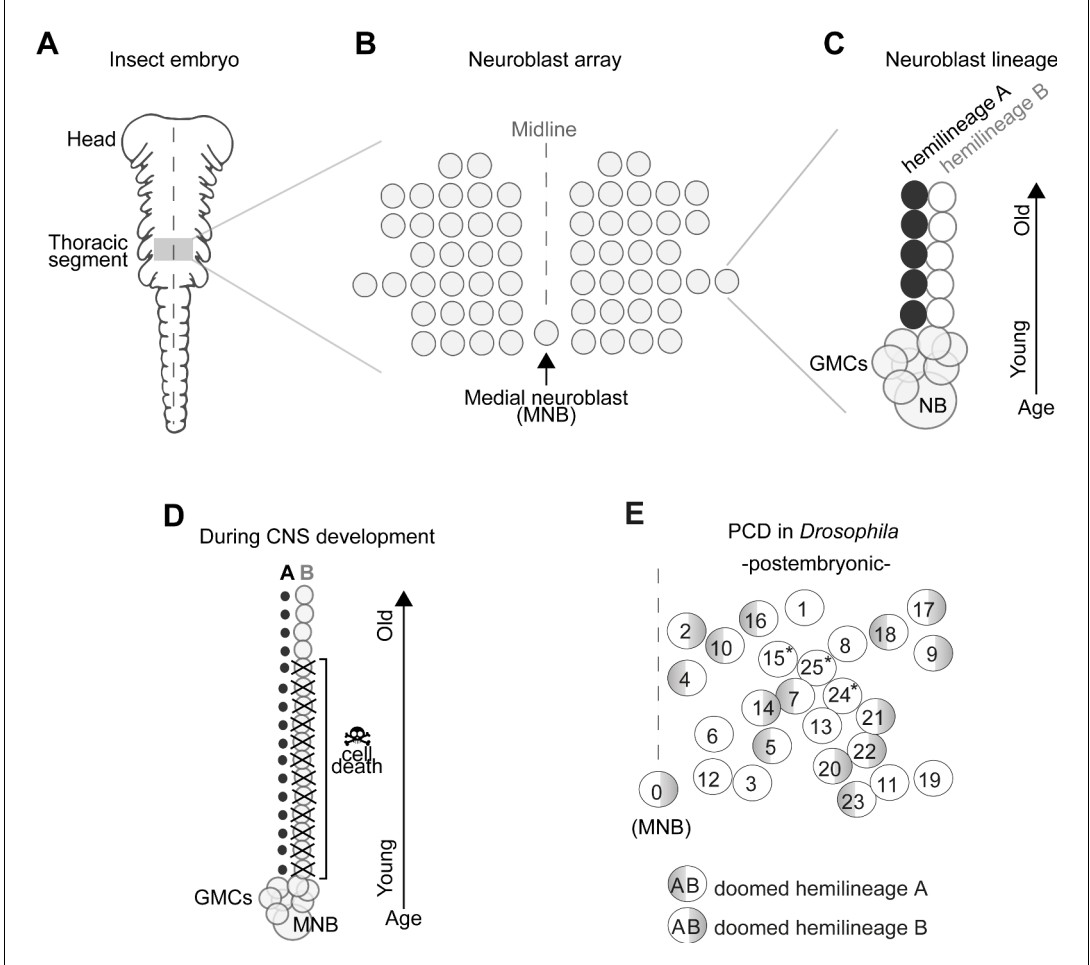

**Figure 1.** Early neurogenesis in insects. (**A**) Cartoon of a young grasshopper embryo, here used as a depiction of a generic insect embryo (modified from *Truman, 1996*). Thoracic territory where one segment-worth of neuroectoderm generates an array of neuroblasts (NB) (grey box). (**B**) Schematic of NB array showing bilaterally symmetric NBs organised in seven rows and six columns and a single median neuroblast (MNB) (modified from *Truman, 1996*). (**C**) Schematic of a lineage derived from an NB. Every NB buds off a ganglion mother cell (GMC) which undergoes a terminal division to generate two neurons with distinct cell fates, an A cell (black) and a B cell (grey). As the A and B cells result from a single division, one cannot be produced without the other. After several rounds of GMC divisions, a lineage produced by a single NB is composed of two half-lineages: 'hemilineage A' made up of all the A cells and 'hemilineage B' made up of B cells. Arrow indicates relative age, with newly born cells located close to the NB. (**D**) Schematic of MNB during development with 'hemilineage A' cells (black) and 'hemilineage B' (grey). The first neurons in hemilineage B cells survive (they are the oldest). After this point, all hemilineage B cells are removed by PCD whereas 'hemilineage A' cells from the same GMC division are left intact. (**E**) Schematic representation of the pattern of hemilineage-specific cell death in one hemisegment in the mesothorax. Each circle represents one lineage produced by one NB. Numbers represent postembryonic lineage nomenclature established by *Truman et al., 2004*. Shaded half circle refers to doomed hemilineage. Lineage 0 is the postembryonic name given to the MNB lineage. Asterisks mark lineages in which one hemilineage produces motor neurons and the other hemilineage generates glia (*Enriquez et al., 2018*; *Lacin and Truman, 2016*).

To mimic such an evolutionary role for PCD, we use the powerful genetic tools available in *Drosophila* to block death in one doomed hemilineage. We chose to target the MNB lineage for the following reasons; Its easy-to-locate position made the MNB identifiable in all developing insects described from as early as 1891 by *Wheeler, 1891*, and spanning all insect orders from wingless silverfish to locusts, beetles, moths and flies (*Bate, 1976*; *Biffar and Stollewerk, 2014*; *Booker and Truman, 1987*; *Doe, 1992*; *Doe and Goodman, 1985*; *Hartenstein and Campos-Ortega, 1984*; *Shepherd and Bate, 1990*; *Tamarelle et al., 1985*; *Truman and Ball, 1998*; *Truman and Bate, 1988*). The MNB gives rise to two distinct populations of neurons, one GABAergic and one octopaminergic, which are also homologous across insects (*Campbell et al., 1995*; *Jia and Siegler, 2002*;

*Lacin et al., 2019*; *Pflüger and Stevenson, 2005*; *Rowell, 1976*; *Siegler and Pankhaniya, 1997*; *Siegler et al., 2001*; *Siegler et al., 1991*; *Stevenson and Spörhase-Eichmann, 1995*; *Thompson and Siegler, 1991*; *Witten and Truman, 1998*). There appears to be a relationship between cell number and function in these populations. Flying insects have greater numbers of octopaminergic neurons within segments that control wings (*Stevenson and Spörhase-Eichmann, 1995*), while grasshoppers have more GABAergic neurons in the fused metathoracic/abdominal ganglia, where they receive auditory input from the abdomen (*Witten and Truman, 1998*; *Thompson and Siegler, 1991*). Alongside differences in numbers of the same cell type between segments and species, numbers of GABAergic and octopaminergic neurons found in one segment are never equal. This is especially intriguing as during development each GABAergic neuron is a sister cell to an octopaminergic neuron, arising from one cell division and are produced in equal number (see *Figure 1C*). The greater number of GABAergic cells in each segment results from PCD targeting octopaminergic neurons in both grasshoppers (*Jia and Siegler, 2002*) and fruit flies (*Truman et al., 2010*) (see *Figure 1D*). Pieced together, these data suggest that, at least in part, the evolution of some behaviours can be explained by variation in the number of octopaminergic neurons caused by PCD during MNB development.

Octopamine release in the thoracic ganglion has been reported to induce and maintain rhythmic behaviours such as stepping movements and flight muscle contractions in locusts (*Sombati and Hoyle, 1984*) and walking, wing flicking and hindleg grooming in decapitated fruit flies (*Yellman et al., 1997*). All octopaminergic neurons produce tyramine as well, the precursor of octopamine, and tyramine has also been shown to induce fictive walking and flight in a thoracic preparation in locusts (*Rillich et al., 2013*). Throughout our work, we do not discriminate between the role of tyramine and octopamine release from hemilineage 0B and collectively refer to these neurons as octopaminergic. Consistent with its role in both (1) walking and (2) flight, we show that (1) blocking PCD in the octopaminergic hemilineage produced by the MNB in *Drosophila melanogaster* results in mature differentiated 'undead' neurons that survive into adulthood, elaborate complex arborisations and induce walking when activated; and (2) PCD may be responsible for reducing hemilineage 0B in the mesothorax of the flightless swift louse *Crataerina pallida*. Alongside, we propose that PCD may have caused reductions in flight hemilineages within thoracic networks in another true fly, *Braula coeca* (the bee louse), during the evolution of flightlessness. Additionally, using new tools in *D. melanogaster*, we demonstrate that PCD takes place in these neurons early, very soon after they are born. We find evidence of this early PCD in primitively wingless firebrats and hippoboscid louseflies suggesting that it is deployed widely. This 'early' death is categorically different to the neuronal death described in the majority of studies in insects, that focus on hormonally gated PCDs occurring at moults (*Pinto-Teixeira et al., 2016*).

Our work highlights the importance of viewing hemilineages as functional units of neurodevelopment in all insects and shows that their alteration through an early mode of PCD can lead to adaptive changes in central circuits during evolution.

## Results

### An early and rapid mode of developmental cell death eliminates significant numbers of newly born neurons throughout postembryonic development in *Drosophila*

First, we wondered what specific type of PCD is responsible for sculpting VNC lineages in *D. melanogaster*, reasoning that only by gaining insight into the exact developmental process involved can we understand its role in nervous system evolution. The majority of studies on neuronal PCD in insects have focused on its role at metamorphic transitions, where death eliminates fully differentiated neurons either at puparium formation (*Truman et al., 1994*) or in adults post-eclosion (*Draizen et al., 1999*; *Kimura and Truman, 1990*). Both of these remodelling events are gated by ecdysteroids. However, our previous observations in *Drosophila* (*Truman et al., 2010*), together with studies in the fly brain (*Nordlander and Edwards, 1968*; *Kumar et al., 2009*; *Lin et al., 2010*; *Lovick et al., 2017*), made us consider that hemilineage-specific PCD takes place early, in newly born neurons. So far, the dynamics of cell death has been difficult to evaluate on a cell-by-cell basis within a complex nervous system.

To interrogate postembryonic neuronal death, we have built a novel genetically encoded effector caspase probe called SR4VH (*Figure 2A,B*). SR4VH consists of a membrane-bound red fluorescent protein (Src::RFP) and a yellow fluorescent protein with a strong nuclear localisation signal from histone H2B (Venus::H2B) separated by four tandem repeats of the amino acid sequence DEVD. When effector caspases cleave the DEVD site, Venus accumulates in the nucleus while RFP remains bound to the cell membrane (*Figure 2B*). This reporter is similar in design to Apoliner (*Bardet et al., 2008*), but has different subcellular localisation signals as well as four tandem caspase cleavage sites instead of one. We also found that tethering the probe to the membrane with the myristoylation signal from Src means that there is no excess signal accumulation in the Golgi apparatus (Mukherjee et al., in preparation). The nuclear localisation signal from H2B allows for highly efficient sequestration of cleaved Venus in the nucleus even in late stages of apoptosis, when the nuclear membrane is likely compromised.

Using the GAL4/UAS system and the NB driver *Worniu-GAL4*, we found we could visualise post-embryonic neurogenesis and label up to 20 of the most recently born progeny from a single NB (this is due to GAL4 and reporter perdurance). The number of progeny we can detect at any one time using *Worniu-GAL4* varies from 10 to 20, most likely as a result of differential proliferation rates across lineages. We confirmed that SR4VH is reliable as a reporter for cell death in larvae by analysing its expression pattern in all lineages of postembryonic neurons in the thoracic VNC and comparing it to our previous work on MARCM homozygous mutant clones of the initiator caspase *Dronc* (*Truman et al., 2010*; *Figure 2C,D,E,F* and *Figure 2—figure supplement 1*). We found dying cells associated with lineages in the brain and VNC throughout the whole of postembryonic neurogenesis (*Figure 2C,D,E,F*), which lasts for 3.5 days, from mid-2nd instar (L2) to 12 hr after pupariation. As previously suggested (*Truman et al., 2010*), the time course of PCD indicates that cells die early – very soon after they are born – often before they have even extended a neuritic process. This death appears to be unlike the 'trophic' PCD found in vertebrates, where a neuron extends a process, interacts with its target cell and dies in the absence of appropriate survival signals. In support of an early onset of PCD, we were able to see sequential stages of cell death, dependent on the distance from the NB (*Figure 2G,H,I* and *Figure 2—figure supplement 1B*). Older cells located further away from the NB appear to be at a more advanced stage of PCD indicated by the complete translocation of Venus from the membrane to the nucleus (Cell three in *Figure 2G,H*) and by the accumulation of RFP-positive dead cell membranes close to the lineage bundle (arrowheads in *Figure 2E,F*).

The number of dying cells within a doomed lineage varied from 1 to 8, with most lineages containing 1–2 dying cells from a total of 10–20 cells labelled with *Worniu-GAL4* (dying cells/lineage: $1.3 \pm 1.5$ given as average $\pm$ standard deviation; n = 444 doomed lineages from 5 VNCs). From a total of 444 doomed lineages, 243 harboured more than one dying cell, of which 148 displayed a progression of cell death (*Figure 2G,H*). *Truman and Bate, 1988* approximated the cell cycle of an NB to 55 min and that of a GMC to 6.5 hr, with 7 GMCs present in a proliferating lineage at all times. Therefore, after subtracting the NB and GMCs from clusters of 10–20 *Worniu-GAL4*-labelled cells, 2–12 will be neurons which resulted from 1 to 6 divisions, each separated in time by 55 min. This means that PCD was initiated early, at some time between 0 and 5.5 hr after neurons were born.

To look at death specifically during the development of the MNB lineage (lineage 0) we imaged SR4VH in wandering L3 larvae and used molecular markers to identify members of hemilineage 0A and 0B. The transcription factors Engrailed/Invected (En/Inv) are known to be expressed in immature and fully differentiated interneurons of hemilineage A (*Allen et al., 2020*; *Lacin et al., 2019*; *Lacin et al., 2014*; *Truman et al., 2004*). As previously reported, the mature differentiated octopaminergic neurons found in hemilineage B express the transcription factor Vestigial (Vg) (*Landgraf et al., 2003*). Here, we find that a small number of immature postembryonic neurons (about 3–5) in close proximity to the MNB also express Vg (*Figure 2J*). Within these immature neurons the expression of Vg and En are mutually exclusive. Using *Worniu-GAL4* to drive SR4VH we found that only the engrailed-negative cells are undergoing apoptosis (*Figure 2K*), i.e. the same small number of cells that express Vg. Their proximity to the MNB suggests that Vg-positive B cells (i.e. immature octopaminergic neurons) undergo an early death, very soon after they are born.

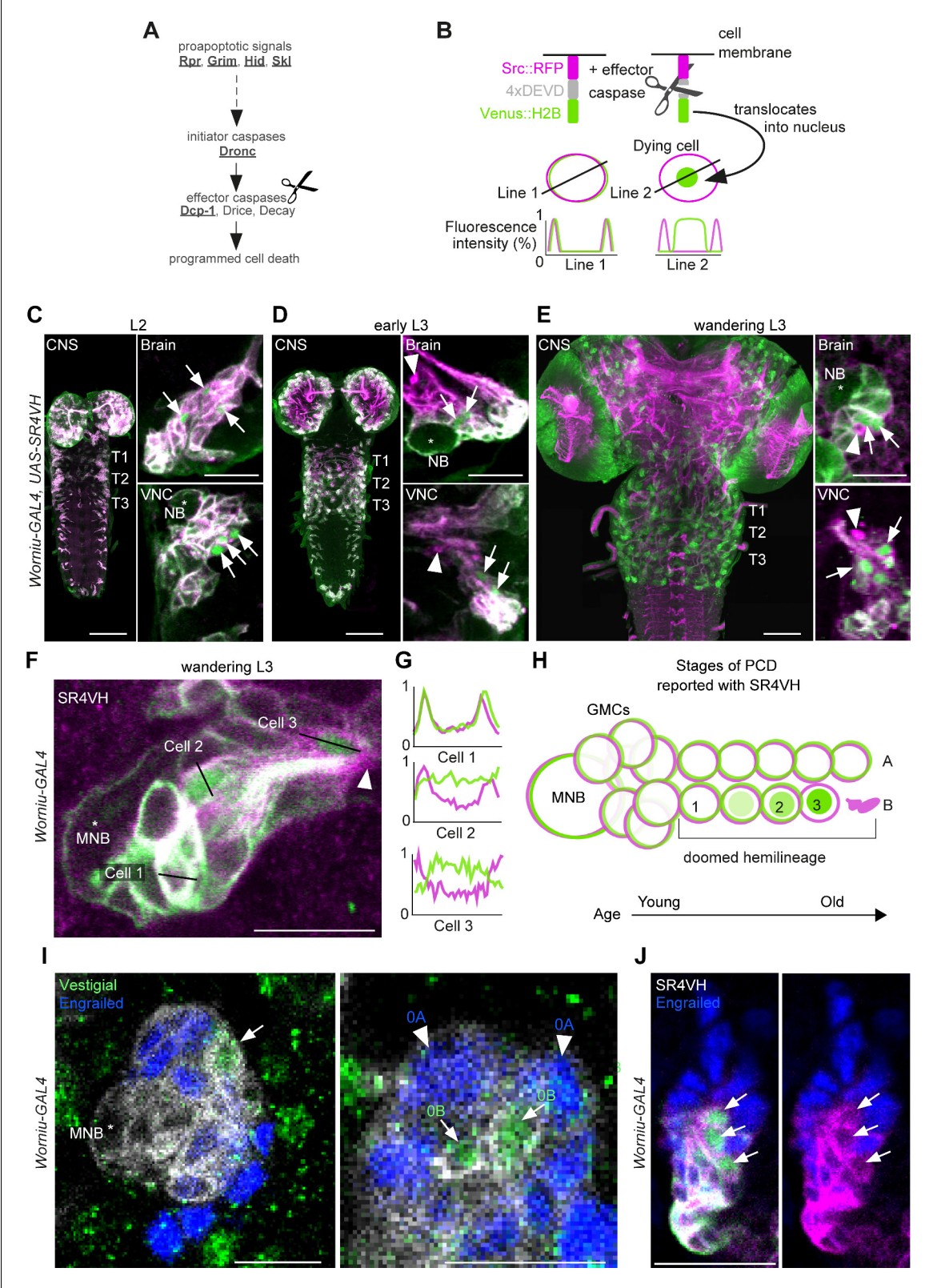

**Figure 2.** Early neuronal cell death occurs throughout postembryonic development. (**A**) Simple schematic of programmed cell death (PCD) in *Drosophila*. Elements disrupted in this study, or used as a PCD readout are underlined. (**B**) Schematic of effector caspase reporter SR4VH (top) and idealised fluorescence patterns in two cells with distinct caspase activity (bottom): RFP (magenta) and Venus (green) present in the cell membrane (Line 1) versus RFP at the cell membrane and Venus accumulation in the nucleus of a cell undergoing PCD (Line 2). (**C–E**) SR4VH driven by *Worniu-GAL4*

*Figure 2 continued on next page*

*Figure 2 continued*

reveals dying cells in the central nervous system (CNS) throughout postembryonic neurogenesis in a 2nd instar (**C**, n = 9), early 3rd instar (**D**, n = 8) and wandering 3rd instar (**E**, n = 18) larva. Right panels show examples of lineages with dying cells located close to the NB (*) in the brain (top) and ventral nerve cord (VNC, bottom). Arrows indicate dying cell, arrowheads indicate RFP-positive dead cell membranes. Scale bars, 50 μm (left panels), 10 μm (right panels). (**F**) SR4VH driven by *Worniu-GAL4* reveals younger cells at earlier stages of cell death (Cell 2) are located closer to the MNB than older cells at later stages of cell death (Cell 3), which are closer to the lineage bundle (arrowhead). Image represents a single optical section. Scale bar, 10 μm. (**G**) Fluorescence intensity profiles (normalised to the maximum value along the line for each channel) plotted along the lines indicated in (**F**). (**H**) Schematic representing successive stages of cell death correlated with distance from NB and cell age in doomed lineages as reported with SR4VH. (**I**) Revealing markers for lineage 0. Immature hemilineage A progeny revealed with anti-Engrailed (blue – arrowheads) and immature hemilineage B progeny labelled with anti-Vestigial (green – arrows) in a third instar *Worniu-GAL4; UAS- CD8::GFP* larva (white). Expression of markers is mutually exclusive. Scale bars, 10 μm. (**J**) SR4VH driven by *Worniu-GAL4* together with antibodies for Engrailed (blue) reveal that only Engrailed-negative cells from hemilineage 0B undergo PCD (white arrowheads) during postembryonic development in the thoracic VNC. Scale bar, 10 μm. n = 6.

The online version of this article includes the following figure supplement(s) for figure 2:

**Figure supplement 1.** SR4VH reveals successive stages of cell death in lineages with doomed hemilineages.

## Blocking PCD in *Drosophila* generates identifiable, differentiated populations of undead neurons

After observing the extent of early PCD during development, we wondered if, by reducing PCD, we could generate novel functional expansions of a hemilineage. To explore this, we made use of the powerful genetic tools available in *Drosophila* to block PCD in the MNB lineage to determine if 'undead' cells survive into adulthood, elaborate their neurites and acquire a distinctive neurotransmitter identity.

From our previous work (*Truman et al., 2010*), we know that during postembryonic neurogenesis MNB hemilineage A survives, expresses Engrailed (*Truman et al., 2004*) and differentiates into GABAergic interneurons (*Lacin et al., 2019*). Because during embryonic development the MNB hemilineage B produces a small number of octopaminergic neurons, we hypothesised that preventing PCD would generate additional octopaminergic neurons in the later postembryonic phase of neurogenesis. In postembryonic nomenclature, all the neurons generated by the MNB are collectively called lineage 0 and therefore we will refer to octopaminergic neurons generated by the MNB as hemilineage 0B.

Using the octopaminergic neuron driver, *TDC2-GAL4,* we observed a 4- to 9-fold increase in the number of octopaminergic neurons in the thoracic VNC of *H99/XR38* adult flies deficient for proapoptotic genes ($hid^{+/-}$, $grim^{+/-}$, $rpr^{-/-}$ and $skl^{+/-}$) (*Peterson et al., 2002*; *White et al., 1994*) compared with wild-type control animals (*Figure 3A,B,C,D*), (T1: 20.9 ± 2.3, Mann-Whitney U = 0, p = 0.0002; T2: 26.3 ± 4.4, Mann-Whitney U = 0, p=0.0004; T3: 27.5 ± 3.5, Mann-Whitney U = 0, p = 0.0004; n = 11 each). These 'undead' neurons also express the vesicular glutamate transporter VGlut (*Figure 3E,F*), just like wild-type octopaminergic neurons (*Greer et al., 2005*).

Ideally, to label and manipulate dying neurons from hemilineage 0B, we require a specific driver line expressed only in the newly born doomed neurons. To test if we could use *TDC2-GAL4* to label and manipulate dying neurons from hemilineage 0B during their development, we performed a timeline of expression in wild-type and *H99/XR38* flies (*Figure 3—figure supplement 1*). Unfortunately, even though undead neurons are generated from L2 onwards in *H99/XR38* flies, the *TDC2-GAL4* is only active in the undead cells days later. Gradually, in pupae, TDC2-GAL4 expression reveals the remaining undead B cells (*Figure 3—figure supplement 1C,D*). We concluded that the TDC2 driver line cannot be used to visualise and manipulate newly born postembryonic 'doomed cells'. Instead, TDC2-GAL4 allowed us to accurately reveal 'undead' hemilineage 0B neurons *but* only in the adult (see cartoon *Figure 3G*).

To ensure sparse labelling and the precise manipulation of only doomed cells from hemilineage 0B, we generated postembryonic *TDC2-GAL4*-expressing MARCM clones homozygous for the loss-of-function allele $Dronc^{ΔA8}$ (in which PCD is inhibited) (*Kondo et al., 2006*; *Truman et al., 2010*). This strategy guarantees that, even though cell death can be rescued in other lineages, it is only within the *TDC2*-positive postembryonic neurons that UAS-based tools are expressed.

Analysis of the projection patterns of undead neurons revealed that they display both common and distinct features compared to their wild-type embryonically born counterparts. Similar to wild-type octopaminergic cells (*Monastirioti et al., 1995*), the undead neurons have cell bodies located

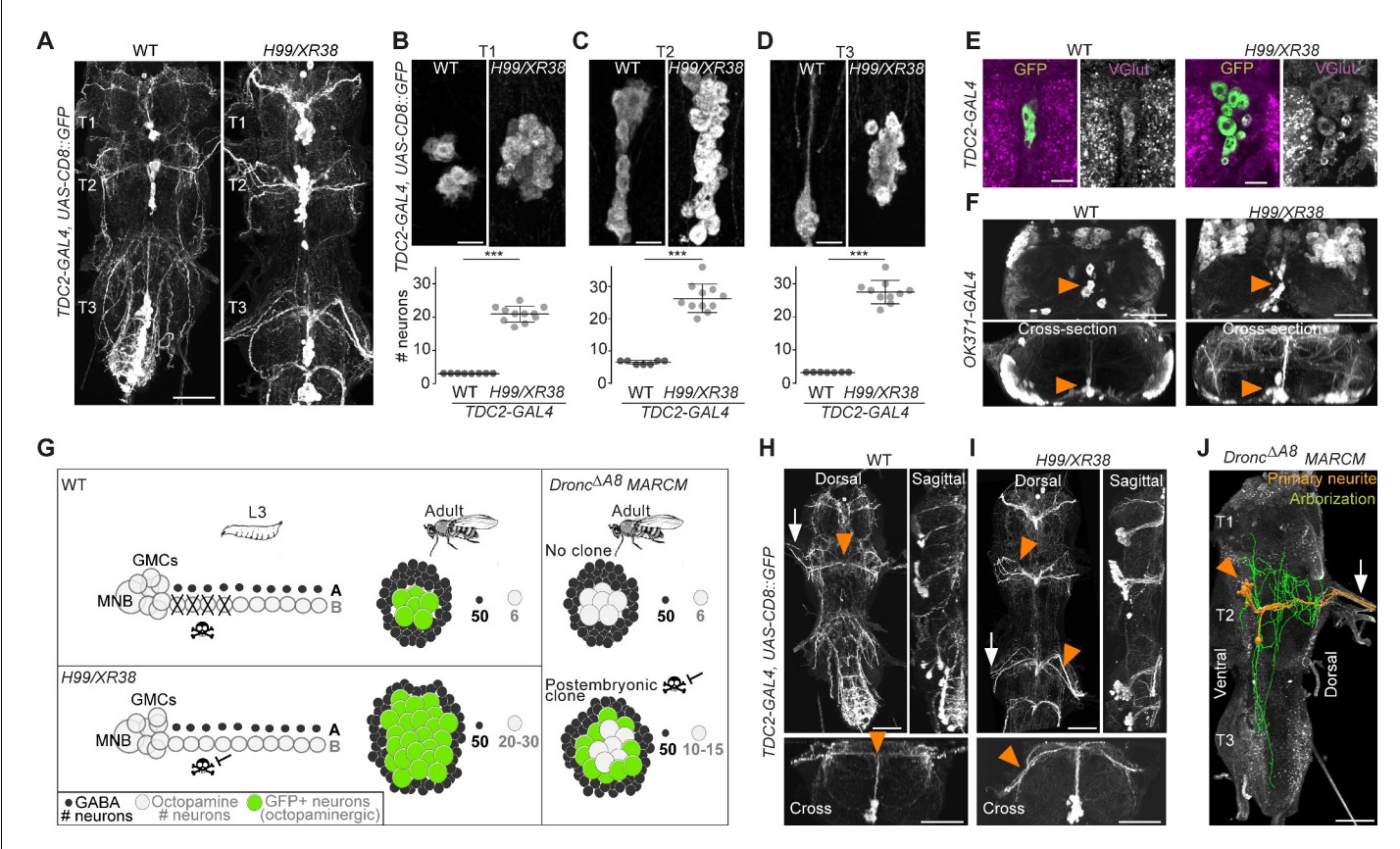

**Figure 3.** Blocking death results in differentiated undead neurons in the medial neuroblast (NB) lineage. (A) CD8::GFP expression driven by *TDC2-GAL4* in octopaminergic neurons from hemilineage 0B in the VNC of wild-type (WT, left) and PCD-blocked adult flies (*H99/XR38* deficient for *hid*[+/-], *grim*[+/-], *rpr*[-/-] and *skl*[+/-], right). Scale bar, 50 μm. (B–D) Quantifications of the number of *TDC2-GAL4*-positive octopaminergic neurons in the VNC of WT and *H99/XR38* adult flies. Bars represent mean ± standard deviation. (B) ***p = 0.0002 in T1, (C) ***p = 0.0004 in T2, (D) ***p = 0.0004 in T3, Mann-Whitney. Scale bar, 10 μm. n = 11 each. Mann-Whitney. Scale bar, 10 μm. n = 11 each. (E) Antibodies for the vesicular glutamate transporter VGlut and (F) GFP expression driven by the glutamatergic driver line *OK371-GAL4* label both WT and undead (*H99/XR38*) octopaminergic neurons. Orange arrowheads indicate cell bodies. Scale bars, 10 μm (VGlut), 50 μm (OK371). (G) Schematic of TDC2-GAL4 expression in postembryonic lineage 0 in wild-type and *H99/XR38* third instar larvae and adults (left panels). Postembryonic hemilineage B populations only start expressing TDC2-GAL4 in early pupal development and maintain it throughout adult life. MARCM mosaic clones that are homozygous for a null *Dronc* allele lack GAL80. These show robust expression of GAL4 in small numbers of surviving postembryonic hemilineage B cells (right panel). In adult WT and *H99/XR38* flies, GFP is expressed in both embryonically born and postembryonic TDC2-positive neurons, while in MARCM flies GFP is *only* present in postembryonic cells. (H–I) CD8::GFP expression driven by *TDC2-GAL4* in WT (G) and *H99/XR38* (H). WT and undead primary neurites project dorsally and branch extensively in the dorsal neuropil. In WT neurons the primary neurite bifurcates at the dorsal midline, while undead neurons are unable to bifurcate and turn to one side (orange arrowheads). In *H99/XR38* flies which contain both WT and undead neurons, the primary neurite to one side is thicker (orange arrowhead). Both WT and undead neurons join thoracic nerves (white arrows). Scale bars, 50 μm. (J) Reconstructed arborisations of undead neurons expressing CD8::GFP driven by *TDC2-GAL4* in flies bearing MARCM clones homozygous for the loss-of-function allele *dronc*[ΔA8] (in which PCD is blocked). The 3D-rendered image is tilted at a 45° angle. Undead neurons have somata that are located at the ventral midline (orange arrowhead), branch extensively in the neuropil (green), have a turning primary neurite (orange) and project to the periphery through a thoracic nerve (arrow). Scale bar, 50 μm.

The online version of this article includes the following figure supplement(s) for figure 3:

**Figure supplement 1.** Postembryonic development of lineage 0B in wild-type and *H99/XR38* flies.
**Figure supplement 2.** Examples of postembryonic MARCM clones in which cell death is blocked.
**Figure supplement 3.** Quantification of undead neuron morphology.

ventrally at the midline, at the posterior border of the thoracic segment (*Figure 3H,I,J*), project a primary neurite in the dorsal-most region of the neuropil, the tectulum (*Court et al., 2020*) and join thoracic nerves (*Pauls et al., 2018*; *Figure 3H,I,J*, *Figure 3—figure supplement 2* and **3**). Unlike wild-type cells which bifurcate and branch extensively in the tectulum, the primary neurite of undead neurons fails to bifurcate, branches in both dorsal and ventral regions of the neuropil and sends

projections to neighbouring segments (*Figure 3J* and *Figure 3—figure supplement 3*). As we describe in *Figure 3—figure supplements 1* and *2, a* few wild-type octopaminergic neurons are produced in all thoracic segments during postembryonic neurogenesis in lineage 0. We propose that the very few bilateral projecting neurons we encounter in our clones are wild-type cells (*Figure 3—figure supplement 2E,F,G*). To avoid any uncertainty when performing our behavioural experiments (below), we excluded flies which contained a bifurcating neuron in undead MARCM clones. Thus, using MARCM clonal approaches we show that undead neurons in hemilineage 0B become octopaminergic, elaborate complex neurites and join thoracic nerves.

## Undead neurons are functional and integrate into motor networks

We next asked if these differentiated undead neurons are functional. To address this, we tested if activating undead neurons with the warm temperature-gated ion channel TrpA1 in headless adult *Drosophila* could elicit behaviours (*Figure 4* and *Video 1*). For this purpose, we deployed the same MARCM-based technique detailed above which ensured that only postembryonic octopaminergic neurons expressed CD8::GFP and TrpA1. The stochastic nature of MARCM allowed for generating both controls and flies with undead neurons in one mating cross using the same genotype, rearing and heat-shock conditions, i.e. there would be animals that would have experienced the heat-shock but have no octopaminergic neurons labelled. This further meant that behavioural experiments were performed blindly and each fly was matched to its control or undead neuron group only following

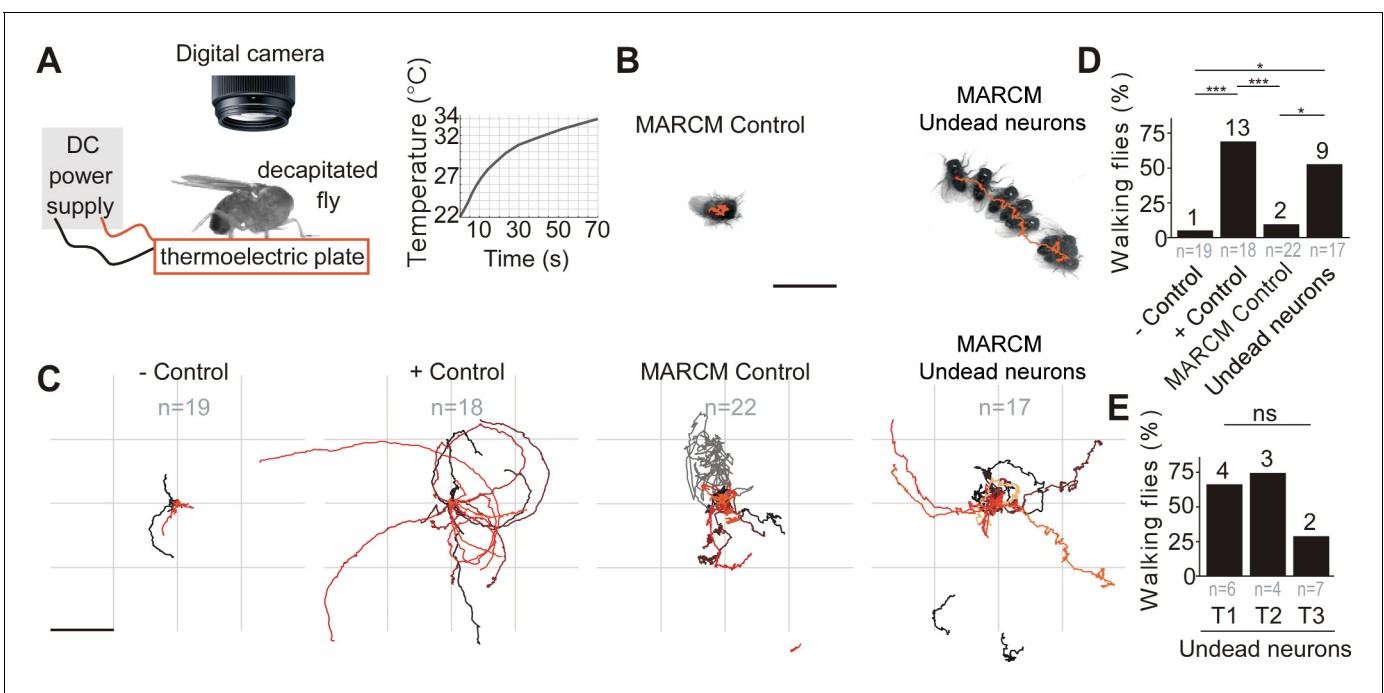

**Figure 4.** Undead neurons are functional: thermogenetic activation of undead neurons induces walking in decapitated *Drosophila*. . (**A**) Schematic of behavioural assay with TrpA1 activation. Decapitated flies are placed on a thermoelectric plate connected to a DC power supply, exposed to a temperature ramp (right panel) and filmed from above using a digital camera. (**B**) Examples of a stationary MARCM Control fly and a walking fly with undead neurons. Images represent maximum intensity projections of 13 frames at 0.3 fps tracing the centroid over time (orange line). Scale bar, 5 mm. (**C**) Fly body tracks generated by identifying the geometric centre of the fly body in each frame and storing the centre coordinates, plotted as a continuous line, one for each fly (walking or stationary) for negative controls (*UAS-TrpA1*), positive controls (*TDC2 >TrpA1*), MARCM control flies and flies with MARCM clones of undead neurons. Each trace represents one individual fly. Scale bar, 5 mm. (**D**) Quantification of the percentage of flies that walked per experimental group. ***p = 0.0002 for negative versus positive control, ***p = 0.0002 for positive controls versus MARCM Control, *p = 0.0242 for MARCM control versus undead neurons, *p = 0.0135 for negative control versus undead neurons, Pearson's chi-squared corrected for multiple comparisons using a Bonferroni correction. n = 19 for negative controls, n = 18 for positive controls, n = 22 for MARCM control, n = 17 for undead neurons. *n* numbers for each group are given below and the number of flies which walked is shown above each bar. (**E**) Quantification of the number of walking undead neuron flies split into three anatomical subgroups according to the location of MARCM clones in T1, T2, or T3. *P*ns = 0.2628, Pearson's chi-squared. n = 6 for T1, n = 4 for T2, n = 7 for T3. Numbers at the base are the number of walking flies. The percentage is shown above each bar.

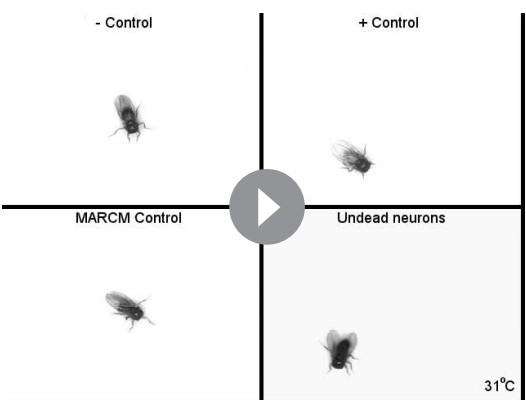

**Video 1.** Video recordings of control and 'undead' decapitated flies during thermogenetic activation. Examples from each behavioural category showing responses to heat-activation in negative control (top left), positive control (top right), MARCM control (bottom left) and undead neurons (bottom right). MARCM control and undead neuron animals were used for extracting the centroid trace provided in *Figure 4B, C*. The increase in temperature is displayed in the bottom right corner. Frames represent recordings from 30 to 70 s.
https://elifesciences.org/articles/59566#video1

dissection and imaging of the VNC. As mentioned above, we excluded flies with MARCM clones containing bilaterally symmetric neurons from our analysis, as these may be wild-type (see *Figure 3—figure supplement 1* and *Figure 3—figure supplement 2E,F,G*). Additionally, we examined the effects of heat exposure in negative control flies with the genotype UAS-TrpA1 and the effects of heat-activation of wild-type octopaminergic neurons in positive controls expressing UAS-TrpA1 driven by TDC2-GAL4.

In *TDC2-GAL4* positive controls (expressing TrpA1 in wild-type embryonic-born octopaminergic neurons), thermogenetic stimulation induced long bouts of locomotion (*Figure 4C,D* and *Video 1*) in 13/18 flies. Importantly, we found *UAS-TrpA1* negative controls (i.e. an absence of a GAL4) and MARCM control flies (containing no GAL4-positive clones; see *Figure 3—figure supplement 2A*) did not walk in response to temperature elevation (*Figure 4B,C, D* and *Video 1*) (1/19 negative control, 2/22 MARCM control). We found that the activation of undead neurons expressing TrpA1 caused decapitated males to walk in 9/17 samples (*Figure 4B,C,D,E* and *Video 1*; also see *Figure 3—figure supplement 2B,C,D* for examples

of MARCM clones of undead neurons) (negative control versus positive control, $\chi^2$ = 17.6, p = 0.0002; negative control versus MARCM control, $\chi^2$ = 0.2 p = 6; negative control versus MARCM undead neurons, $\chi^2$ = 10.2, p = 0.0135; MARCM control versus positive control, $\chi^2$ = 16.8, p = 0.0002; MARCM control versus MARCM undead neurons, $\chi^2$ = 9.1, p = 0.0242; MARCM undead neurons versus positive control, $\chi^2$ = 1.4, p = 1.4283; All comparisons were performed using Pearson's chi-squared and p values were adjusted using a Bonferroni correction). A further analysis of the occurrence of walking after splitting the MARCM undead neuron group into the three anatomical subgroups T1, T2 and T3 according to the location of undead neurons in the pro-, meso- or meta-thoracic segment, yielded no significant differences: 4/6 T1, 3/4 T2 and 2/7 T3 ($\chi^2$ = 2.9, p = 0.262855, Pearson's chi-squared) (*Figure 4E*).

As previously reported, decapitated flies walked slowly by moving their limbs in a seemingly erratic manner, without having a tripod gait (*Harris et al., 2015*; *Yellman et al., 1997*). Flies were considered to be walking if they covered a distance of at least one body length during recordings and if they moved their legs in the order T3-T2-T1 at least once on each side, as evidence of intersegmental coordination (*Strauss and Heisenberg, 1990*). The direction of walking was either forward, sideways or backwards and most flies turned or walked in circles (*Figure 4C*), probably caused by variation in step size (*Yellman et al., 1997*; *Harris et al., 2015*).

Our data are consistent with the observation that octopamine applied to the exposed anterior notum of decapitated flies causes walking (*Yellman et al., 1997*) and suggests that undead neurons are functional and capable of releasing neurotransmitters in the CNS. The extent of walking was greater in positive control flies than in the undead MARCM condition (compare panels in *Figure 4C* and *Video 1*). This is likely because, alongside activating thoracic octopaminergic neurons in the VNC, in the positive controls we also stimulate the severed axons of octopaminergic cells in the brain which send descending projections to the VNC. These are not present in our *TDC2-GAL4* MARCM flies.

To determine if undead neurons are integrated into thoracic motor circuits, we recorded the activity of mixed undead and wild-type octopaminergic neuron populations expressing GCaMP6s (an activity reporter) and tdTomato (an anatomical fiduciary) in intact *H99/XR38* flies during tethered behaviour on a spherical treadmill (*Chen et al., 2018*; *Figure 5A,B*). The complexity of our calcium

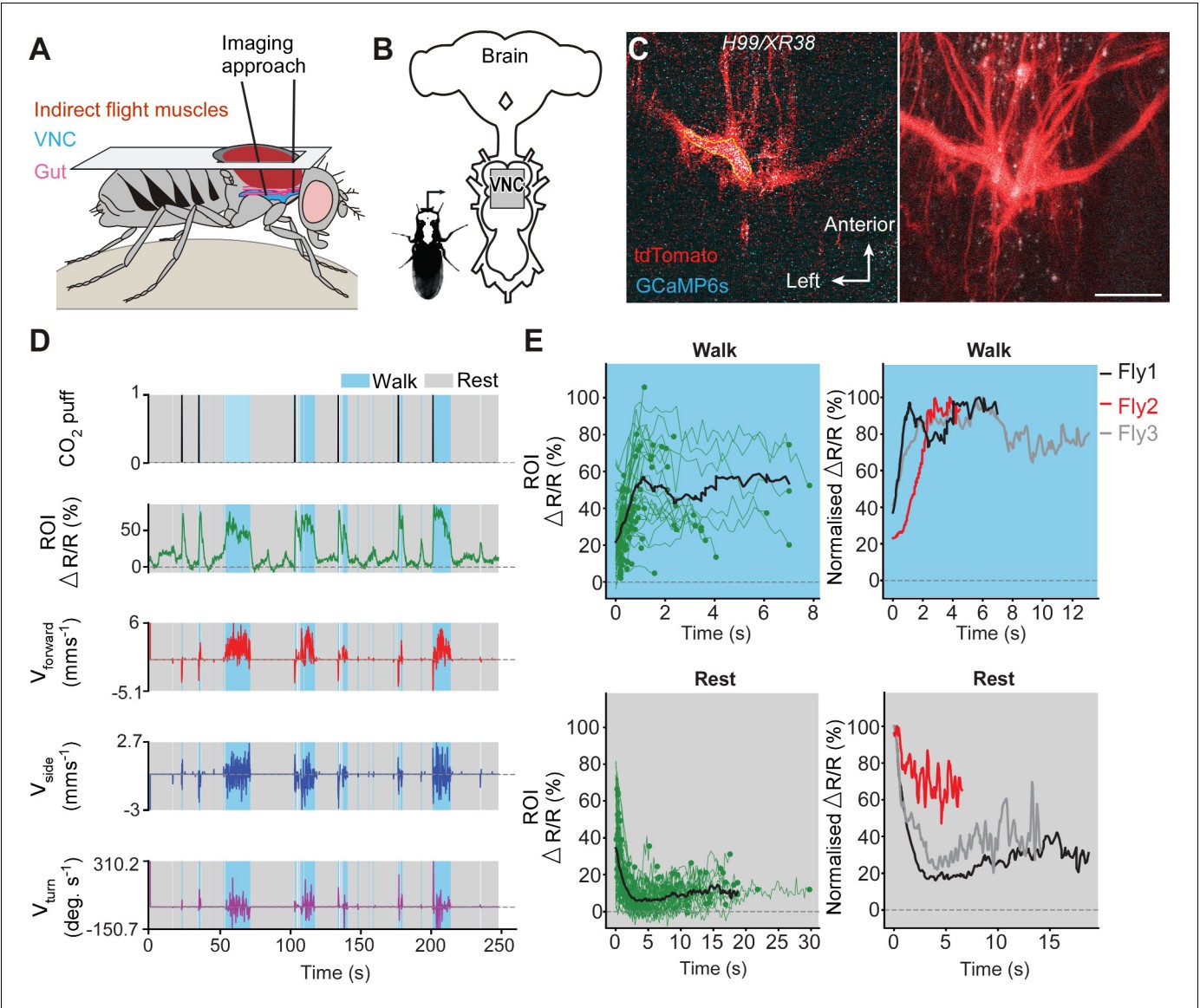

**Figure 5.** Undead neurons integrate into VNC networks: undead neurons are active during naturalistic behaviours in intact adult *Drosophila*. (A) Schematic of the dorsal thoracic dissection and approach for ventral nerve cord functional imaging in tethered, adult flies. (B) Location of the imaging region-of-interest (grey box) with respect to a schematic of the adult CNS. (C) Raw 2-photon image of *TDC2-GAL4-positive* neurons co-expressing tdTomato (red) and GCaMP6s (cyan) in *H99/XR38* flies (left). Region-of-interest used to calculate %ΔR/R is outlined (yellow). Standard deviation z-projection of a dorsal-ventral image stack of the functional imaging region-of-interest in (b) (right). Scale bar, 50 μm. (D) Representative behavioural and functional imaging data in *H99/XR38* flies. Shown are: $CO_2$ stimulation (black), %ΔR/R (ratio of GCaMP6s/tdTomato) signal (green) and ball rotations indicating forward walking (red), sideways walking (blue), and turning (purple). The behaviour of the fly was classified as either walking (light blue), or resting (grey) by applying a threshold on ball rotation speed. (E) (left) Individual (green) and average (black) %ΔR/R traces within each behavioural epoch for walking (n = 82) and resting (n = 86) events processed from 750 s of imaging data. Solid green circles indicate the end of a behavioural epoch. The average trace (black line) was calculated for only periods with four or more traces. (right) Normalised average %ΔR/R traces for three different flies during walking and resting. The average (black) trace is the same as in the left panel.

The online version of this article includes the following figure supplement(s) for figure 5:

**Figure supplement 1.** Wild-type octopaminergic neurons are active during walking in intact adult *Drosophila*.
**Figure supplement 2.** Subregion analysis of calcium signals along the width of a primary neurite in a *H99/XR38* animal.

imaging experiment (see *Figure 5A*), together with a rate of success for obtaining MARCM clones with undead neurons of 15%, prompted us to approach our question whether undead neurons are functional during natural walking by using *H99/XR38* flies. Keeping in mind that, instead of bifurcating, undead neurons collectively take a turn (see *Figure 3I,J* and *Figure 3—figure supplement 2B, C,D*), we interpret activity from the thickest bundle in the bifurcation as belonging to both undead and wild-type neurons. In these animals we observed conspicuous increases in neural activity during air-puff-induced walking in both wild-type controls (*Figure 5—figure supplement 1*) and in *H99/XR38* flies (*Figure 5C,D,E* and *Videos 2*, *3* and *4*). Because undead neurons outnumber their wild-type counterparts by a ratio of 6.5 to 1 (see *Figure 3A,B,C,D*), these results imply that both neuronal types are active in *H99/XR38* flies. Supporting this, we observed an increase in GCaMP6s fluorescence across all subregions along the width of the thickest primary neurite bundle in the bifurcation, which contains all undead neurons together with three wild-type cells (*Figure 5—figure supplement 2* and *Video 4*). Taken together, these data reveal that 'undead' neurons in the adult fly are functional and can integrate into motor networks.

## Hemilineage-specific cell death in the MNB lineage correlates with loss of flight in the swift lousefly *Crataerina pallida*

Our observation that undead neurons functionally integrate into the CNS of adult flies strongly supports the possibility that PCD could be leveraged to modify neural circuits over the course of evolution. Alongside walking, octopaminergic neurons in the MNB lineage have a well-known function in flight (*Roeder, 2005*), and differences in neuron numbers correlate well with varying degrees of flight performance (*Figure 6*). Proficient fliers such as locusts and most flies have more octopaminergic neurons in winged thoracic segments (highlighted in yellow in *Figure 6*), while clumsy fliers such as cockroaches and crickets have similar numbers of neurons across thoracic segments. *Figure 6* reviews our current knowledge of the number of GABAergic (hemilineage A) and octopaminergic (hemilineage B) neurons in the MNB of insects compiled from multiple studies spanning decades of research (see references in *Figure 6*). The lack of data for one population or the other is indicated with a question mark. Alongside our own data from swift louseflies (see *Figure 7*), here we also include our unpublished observations in the horse lousefly (for 0B) and the bee lousefly (for 0A). Having limited samples and antibody, we successfully labelled one preparation each and therefore resort to depicting these as a cartoon in *Figure 6*.

Because octopaminergic population size reflects flight ability across insects, and dipterans generally display larger populations in the mesothorax, we wondered if flies which have lost flight during evolution show reduced numbers of octopaminergic neurons in this segment. To this end, we described the MNB lineage in the flightless dipteran *Crataerina pallida*, the swift lousefly (*Figure 7A*), a viviparous haematophagous ectoparasite of the swift *Apus apus* (*Bequaert, 1952*; *Hagan, 1951*; *Hutson, 1984*; *Walker and Rotherham, 2010a*; *Walker and Rotherham, 2010b*). We labelled lousefly octopaminergic neurons from hemilineage 0B using antibodies for tyramine β-hydroxylase (*Monastirioti et al., 1996*) and by comparing the ratio of octopaminergic cells in the mesothorax (winged segment) and the prothorax (lacks wings), we found that, unlike flying dipterans, the swift lousefly has lost segment-specific variability of cell numbers (fruit fly [2.2 ± 0.2, n = 7] versus swift lousefly [1.1 ± 0.2, n = 3], p=0.012, Mann-Whitney U = 0, Mann-Whitney t-test) (*Figure 7B,C,D,E,F*). In the sister hemilineage 0A we found a considerably larger number of GABAergic neurons (*Figure 7F*), suggesting that PCD may be responsible for the selective elimination of the octopaminergic hemilineage.

We next wondered if we can find evidence of early PCD in hemilineages in the swift lousefly. Swift louseflies are viviparous, with only one

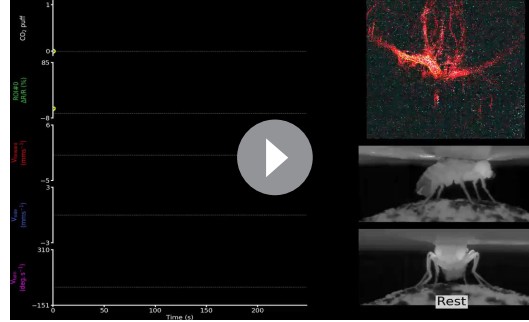

**Video 2.** Recording of 2-photon calcium imaging in undead neurons. Synchronised front and side camera behaviour videography (bottom right), and 2-photon imaging data (top right) used for the data analysis provided in *Figure 5C,D,E*.

https://elifesciences.org/articles/59566#video2

progeny being produced and carried by the female at any one time (see *Figure 7A*, lower panel). Larvae hatch and remain inside the uterus for their entire larval life, feeding on lipid-rich secretions from milk glands until pupariation, when they are deposited in the swift nests and pupate. Adults only emerge the following spring, when swifts return from North Africa (*Hagan, 1951*; *Bequaert, 1952*). To capture postembryonic development, we dissected both larvae from inside female abdomens and pupae staged from Day 0 onwards, indicating days passed since laying (*Figure 7G*). Similar to the tsetse fly (*Truman, 1990*), neurodevelopment is significantly delayed compared to 'typical' dipteran flies - the nervous system only acquires dipteran larval-like features many days after pupariation (*Figure 7H*). Using EdU to label proliferating cells and immunostaining for active Dcp-1 (*Figure 7H,I,J*), we found dying cells located close to NBs throughout the 24 days of pupal neurogenesis (Pop et al., in preparation). In the MNB lineage, which is easily identified by its medial position and projection pattern in the neuropil, we found cell death in thoracic segments at all time points examined, from Day 4 after pupariation to Day 23 (*Figure 7I,J*).

Because cell death is present during neurogenesis in winged insects: selectively eliminating hemilineages in the fruit fly (*Truman et al., 2010*), killing off immature octopaminergic neurons produced by the MNB in the grasshopper (*Jia and Siegler, 2002*), appearing to sculpt neural networks in swift louseflies (see *Figure 7H,I,J*); we wondered if PCD also occurs during CNS development in a 'primitive' wingless insect. Using TUNEL labelling in the firebrat *Thermobia domestica* (*Figure 8A,B*), we found dying cells close to many NBs in all thoracic neuromeres at 50–55% of embryonic development (*Figure 8C,D,E*).

Similar to what we see in *Drosophila* (see *Figure 2* and *Figure 2—figure supplement 1*), our observation that dying cells are found close to NBs in firebrats and louseflies suggests that this early PCD may be a universal and ancestral feature that sculpts the nervous system of all insects. To further explore if changes in PCD may have been deployed during evolution to accommodate adaptive modifications to behaviour, we next searched for evidence of increased PCD in other flight hemilineages of flightless dipterans.

## Increased hemilineage-specific PCD in flightless dipterans may be responsible for adaptive modifications to neural circuits

To explore the possibility that changes in the pattern and/or extent of PCD are adaptive, we looked for evidence of evolutionary modifications in the VNC of yet another species, the bee lousefly *Braula coeca* (*Figure 9A,B,C,D*). *Braula*, a close relative of drosophilids, is wingless, lacks halteres and has an extremely reduced thorax (*Figure 9A*). Bee louseflies spend their entire adult life as kleptoparasites on the honeybee *Apis mellifera* (*Imms, 1942*; *McAlister, 2018*). We specifically asked whether lineages known to function in flight circuitry might be modified in flightless insects. In *Drosophila*, a thorough anatomical study recently described the pattern of innervation for each lineage into known functional domains of the adult neuropil and categorised them accordingly as being involved in leg, wing and both leg and wing control (*Shepherd et al., 2019*). In addition, the functional role of most hemilineages was previously assessed by thermogenetic activation in headless flies and those involved in flight-associated behaviours, such as wing wave, wing buzz or take-off, were identified (*Harris et al., 2015*). Together, these studies provide an excellent starting point for anatomical comparisons of homologous hemilineages which may have served an ancestral role in flight.

Using antibodies for Neuroglian, we compared homologous hemilineages involved in the flight circuits of the two flightless dipterans, the swift lousefly *Crataerina pallida* and the bee lousefly *Braula coeca*. Comparing the Neuroglian-labelled axon bundle width of homologous hemilineages between flightless and flying species, we found that hemilineages 3B, 7B, 11B and 12A, which produce wing waving, wing buzzing or take-off in the fruit fly (*Harris et al., 2015*), hemilineages 6A and 19B, which innervate the wing neuropil, and 5B, which innervates both leg and wing neuropils (*Shepherd et al., 2019*), are reduced in bee louseflies, but not in swift louseflies (*Figure 9B,C,D*). Among the latter, hemilineages 5B and 6A are known to contribute to leg movements and changes in posture, while the role for 19B is yet to be determined (*Harris et al., 2015*). Importantly, 3B, 6A, 11B, 12A and 19B belong to lineages in which both hemilineages survive in fruit flies (see schematic in *Figure 1E*). A difference in axon bundle diameter between sister hemilineages could indicate a difference in cell number, possibly established by PCD during development. We chose to quantify the ratio of axon bundle diameters in lineage three because we expected a reduction in hemilineage 3B, which innervates the wing neuropil, but no change in hemilineage 3A, which projects into the

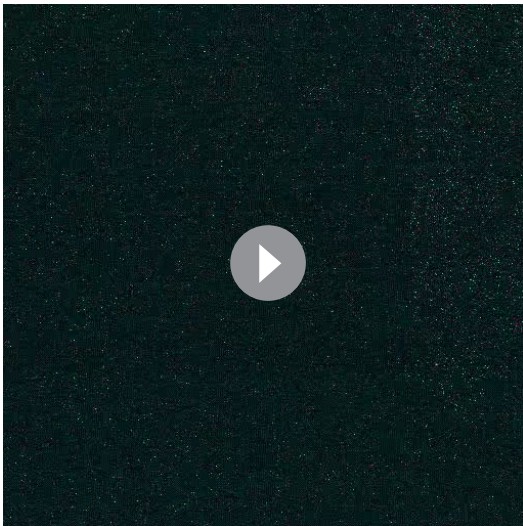

**Video 3.** Z-stack of the imaging area for GCaMP6s activity in undead neurons. A videography showing the imaging plane at different depths of the prothoracic segment corresponding to *Figure 5C*. The thicker left branch likely includes mostly undead neurons. https://elifesciences.org/articles/59566#video3

leg neuropil (*Shepherd et al., 2019*). The fibre tracts of 3A and 3B originate as a common bundle and split only in the intermediate neuropil. After they split, the sister fibre tracts sit in the same plane before they defasciculate, making them easy to trace and compare (*Figure 9E,F,G*). We found that the ratio between sister hemilineages A and B is significantly higher in bee louseflies compared to fruit flies, indicating that hemilineage 3B, which controls flight-related behaviours, is severely reduced in these flightless flies (T1: fruit fly [0.6 ± 0.1, n = 7] versus bee louse [1.8 ± 0.4, n = 8], p < 0.0001, t = −8.084, independent samples t-test; T2: fruit fly [0.7 ± 0.1, n = 7] versus bee louse [1.7 ± 0.4, n = 8], p=0.0001, F = 42.22, Welch's t-test; T3: fruit fly [1.1 ± 0.4, n = 7] versus bee louse [1.7 ± 0.2, n = 8], p = 0.0044, F = 14.84, Welch's t-test). Even though we cannot make a precise inference of cell numbers in each hemilineage, as the fibre tract of hemilineage 3B appears frayed while 3A is more compact in both species, our results clearly show that 3B is greatly reduced, associated with the loss of flight machinery.

## Discussion

### Hemilineage-specific cell death occurs in newly born neurons

To help us understand more about the patterning of PCD, we designed and used a new effector caspase probe SR4VH, which allowed us to interrogate the extent and dynamics of hemilineage-specific cell death. It shows us that an early onset PCD is responsible for the elimination of postembryonic neurons in the fly VNC, that this happens throughout the entire 3.5 days of postembryonic neurogenesis, and is hemilineage-specific. Although PCD has been reported as a fate within lineages in the embryo (*Karcavich and Doe, 2005*; *Rogulja-Ortmann et al., 2007*), the impact of this 'early' and hemilineage-specific PCD on the construction of the adult network has yet to be fully appreciated. This type of PCD is responsible for removing almost half of all postembryonic neurons that are born in the fly (*Truman et al., 2010*). Until now, the most frequently reported type of neuronal death described in insects has been the hormonally regulated PCD that removes mature neurons during the narrow developmental windows at the beginning of metamorphosis and within the first day after adult eclosion (*Draizen et al., 1999*; *Lee et al., 2013*). We now know from our data that these make up only a small fraction, compared to the total number of neuronal deaths in the fly.

Our SR4VH probe allows us to see that newly born neurons initiate cell death within the first 5.5 hr after birth. We can capture different stages of cell death: with young cells at very initial stages of PCD located closer to the NB, while older cells at more advanced stages of PCD with RFP labelled cell membranes found close to the lineage bundle. Importantly, death happens before neurites have extended,

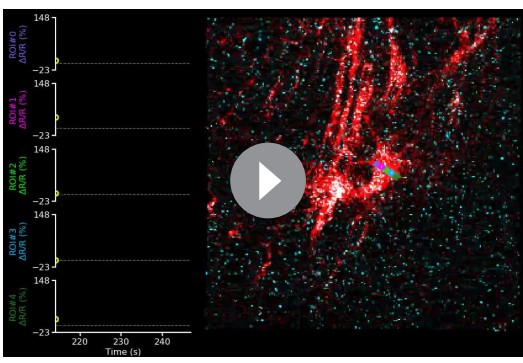

**Video 4.** Subregion neuronal activity patterns during walking. Imaging data used for analysis in *Figure 5—figure supplement 2*. Shown are %ΔR/R traces for R. https://elifesciences.org/articles/59566#video4

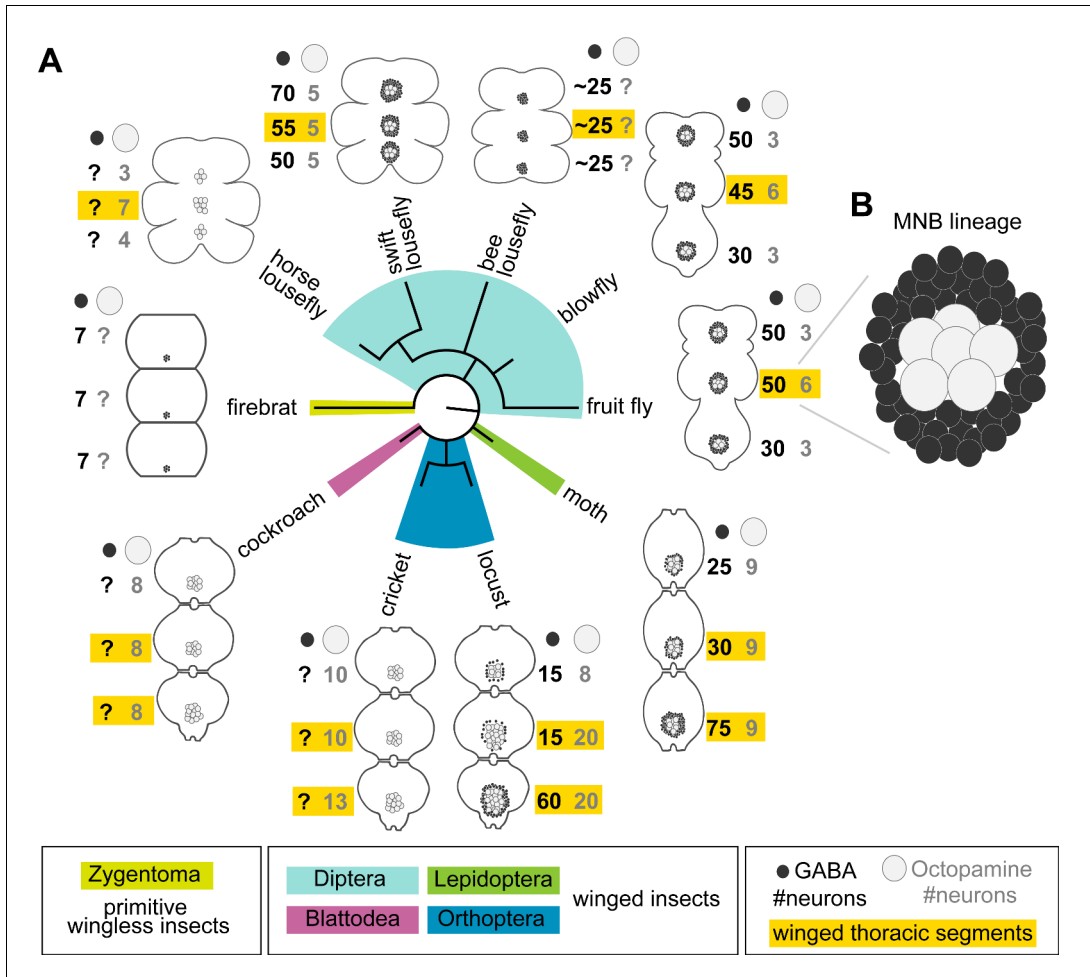

**Figure 6.** Median neuron numbers vary between insect species. (**A**) Schematic showing summary of thoracic midline neuron population data from 5 orders of insects (colour-coded in the phylogenetic tree and in the figure key bellow) including the primitive wingless firebrat *Thermobia domestica*, the cockroach *Periplaneta americana*, the cricket *Gryllus bimaculatus*, the locust *Schistocerca gregaria*, the moth *Manduca sexta*, the fruit fly *Drosophila melanogaster*, the blowfly *Calliphora vicina*, the horse lousefly *Hippobosca equina* and the flightless bee lousefly *Braula coeca* and swift lousefly *Crataerina pallida*. The numbers of GABAergic neurons (black cells) and octopaminergic neurons (grey cells) produced by the MNB is given for each thoracic segment. Except for the moth, a higher number of octopaminergic neurons can be found in winged segments in flying insects (yellow boxes). Cell numbers in this homologous lineage vary both between segments and species. Data on firebrats, cockroaches, crickets, locusts, moths, fruit flies and blowflies are compiled from ***Monastirioti et al., 1995***; ***Stevenson and Spörhase-Eichmann, 1995***; ***Witten and Truman, 1998***; ***Schlurmann and Hausen, 2003***; ***Lacin et al., 2019***; and unpublished data from Dacks, Pflüger and Hildebrand (AM Dacks, personal communication, May 2020), while data on horse, swift and bee louseflies are from our own work. (**B**) Cartoon of *Drosophila* mesothoracic midline lineage populations with 'hemilineage A' cells revealed by GABA immunoreactivity (black) and 'hemilineage B' revealed by octopamine immunoreactivity (grey).

strongly suggesting that this PCD is not an analogue of neurotrophic death, found in vertebrates - where neuron-target interactions play a major role in the decisions of cell survival (***Dekkers and Barde, 2013***). In *Thermobia domestica* and *Crataerina pallida*, where the use of sophisticated genetic reporters such as SR4VH was not yet possible, EdU incorporation to mark dividing cells and immunolabelling for the active effector caspase Dcp-1 as a proxy for cell death revealed dying cells close to sites of division. Dying cells were found in the proximity of NBs (identifiable in all insects by their large size and position in the outermost layer of the CNS cortex) and far removed from mature neurons which congregate in the innermost layer of the cortex, adjacent to the neuropil. Therefore,

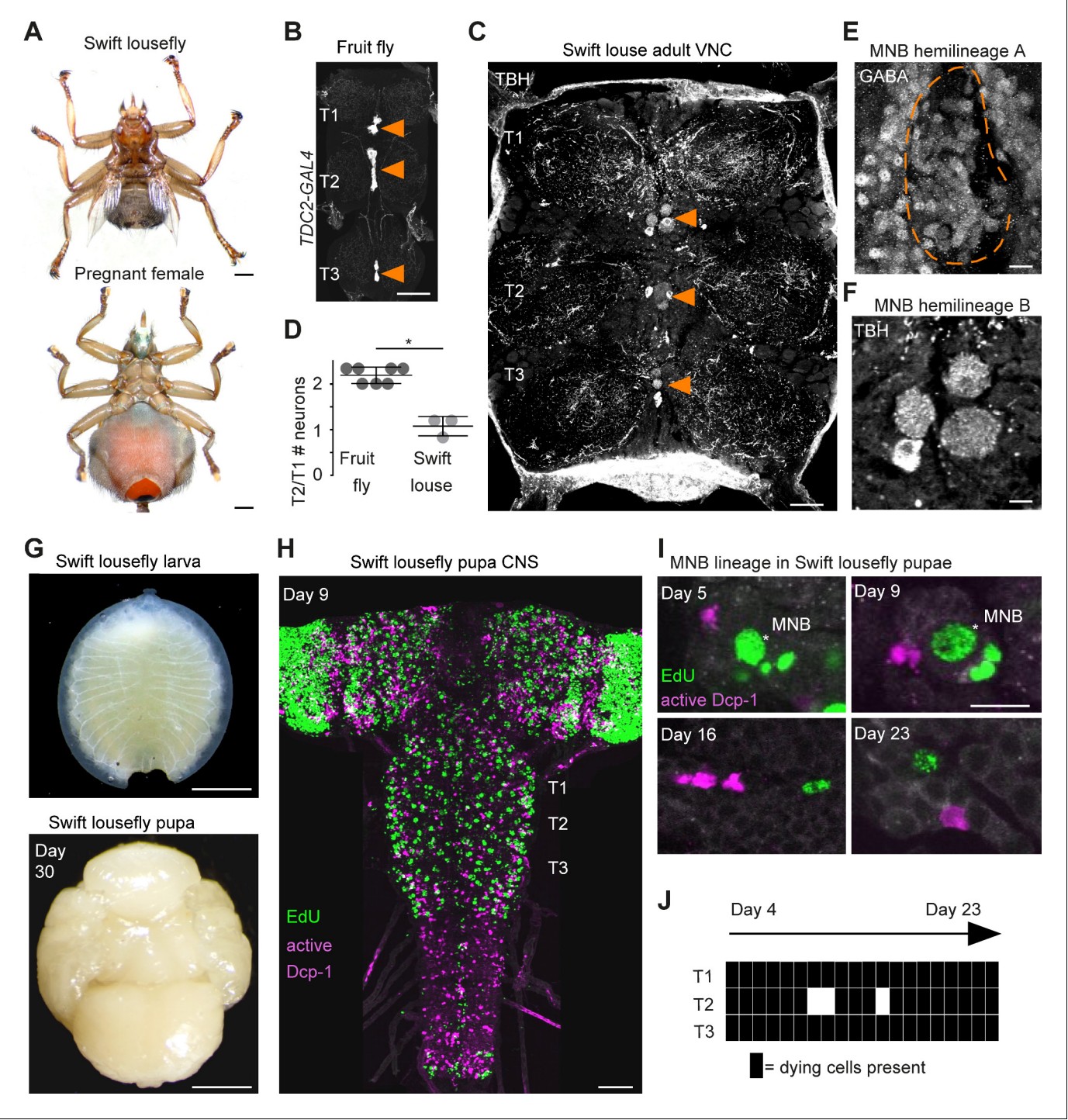

**Figure 7.** Hemilineage-specific cell death may be responsible for reduced octopaminergic neurons in the swift lousefly. (**A**) Dorsal view of an adult swift lousefly with vestigial wings (top). Ventral view of a female pregnant with a prepupa (bottom). Scale bars, 1 mm. (**B**) Wild-type octopaminergic neurons in hemilineage 0B in a *Drosophila melanogaster* VNC labelled with CD8::GFP driven by *TDC2-GAL4* (orange arrowheads). Scale bar, 50 µm. (**C**) Octopaminergic neurons in hemilineage 0B in a swift lousefly VNC labelled with antibodies for tyramine β-hydroxylase (TBH, orange arrowheads). Fluorescence in the neuropil is derived from secondary antibodies trapped in the tracheal system and does not mark the true presence of TBH protein. Scale bar, 50 µm. (**D**) Quantification of T2/T1 number of octopaminergic neurons in fruit flies and swift louseflies shows that swift louseflies have lost the T2-specific higher numbers typical of flying dipterans (*p = 0.012, Mann-Whitney. n = 7 fruit flies, n = 3 swift louseflies). Bars represent mean ± standard deviation. (**E**) Cluster of cell bodies belonging to hemilineage 0A (dashed outline) labelled with antibodies for GABA and (**F**) cell bodies belonging to hemilineage 0B labelled with TBH antibodies in the prothorax (**T1**) of a swift lousefly. (**G**) Swift lousefly larva (top) and Day 30 swift lousefly pupa

*Figure 7 continued on next page*

*Figure 7 continued*

removed from its puparial case (bottom). Scale bars, 1 mm (**H**) EdU labels proliferating cells and antibody-labelling for active Dcp-1 reveals dying cells in the CNS of a swift lousefly pupa 9 days after pupariation. Scale bar, 50 µm. (**I**) Dying cells in lineage 0 labelled with antibodies for active Dcp-1 are located close to proliferating cells (e.g., NB*) throughout neurogenesis at Day 5 (top left), Day 9 (top right), Day 16 (bottom left) and Day 23 (bottom right) after pupariation. Scale bar, 10 µm. n = 1 each. (**J**) The occurrence of active Dcp-1-positive cells in lineage 0 in T1, T2 and T3 from Day 4 to Day 23 after pupariation in swift lousefly pupae (n = 1 each). Each black box indicates one occurrence.

The online version of this article includes the following source data for figure 7:

**Source data 1.** Quantification of T2/T1 number of octopaminergic neurons in fruit flies and swift louseflies.

we speculate that these cells are immature neurons and propose that they too undergo a death with rapid onset following division, similar to the early hemilineage-specific cell death we see with SR4VH in fruit flies.

The critical question that these data bring into focus is how early onset PCD is orchestrated, especially that an early intrinsically determined mode of cell death seems to be widespread across animals, from *C. elegans* to mice (*Fricker et al., 2018*; *Southwell et al., 2012*). Early PCD likely involves a combination of intrinsic patterning and cell-cell interactions between sibling neurons which ultimately deploy the activity of proapoptotic genes *rpr*, *hid*, *grim* and/or *skl*. Previously, patterning genes such as *Ubx* have been shown to contribute to the survival of hemilineages in the thoracic VNC in a parasegment-specific manner (*Marin et al., 2012*), while the transcription factor Unc-4 has been recently demonstrated to provide neural identity to hemilineages involved in flight (*Lacin et al., 2020*). Such spatial patterning is also required to establish NB identity which, in turn, can determine which hemilineage is maintained and which dies. In the developing optic lobe *Bertet et al., 2014* have shown that the temporal sequence of transcription factors expressed in NBs (inherited by the GMC and newly born neurons) produces a switch in the selective survival of one hemilineage over the other. In this manner, the changes to NB identity which have been documented in other insects (*Biffar and Stollewerk, 2014*), despite a conserved NB array and progeny, could in turn influence the pattern of PCD. Alongside, cell-cell interactions between newly born neurons could influence fate choices. The requirement of interactions between newly born siblings in determining asymmetric fates has been shown in the grasshopper VNC (*Doe et al., 1985*;

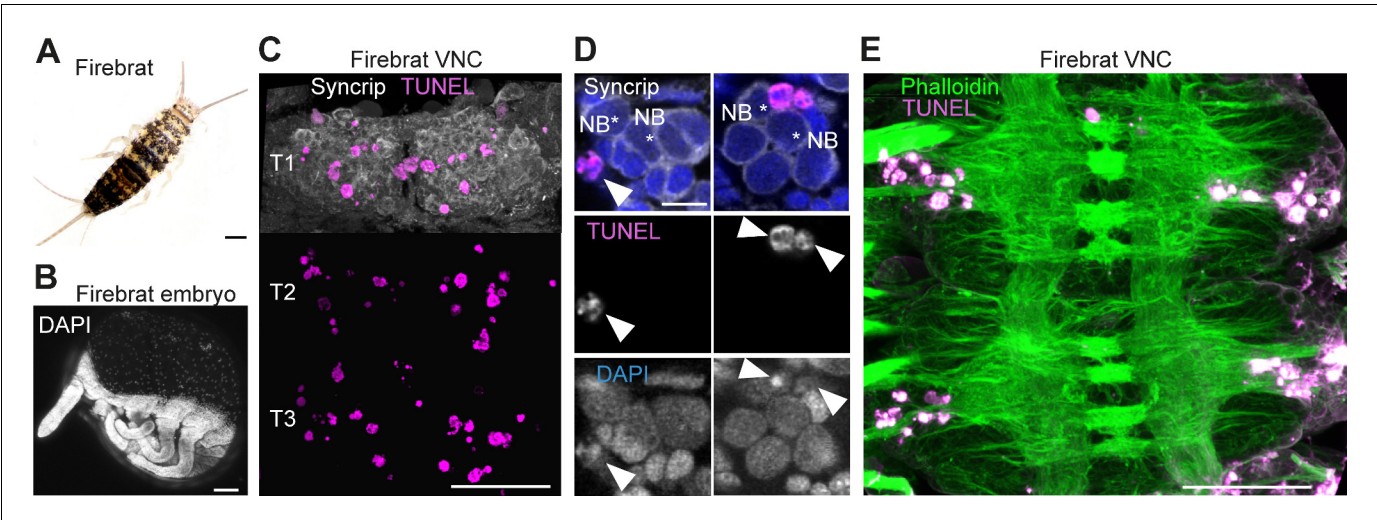

**Figure 8.** Cell death during neurogenesis in the 'primitive' wingless firebrat. (**A**) Adult firebrat. Scale bar, 1 mm. (**B**) Maximum intensity projection of DAPI staining in a wholemount firebrat embryo (*Thermobia domestica*) at 50–55% of embryonic development. Scale bar, 100 µm. (**C**) Dying cells in the thoracic VNC of a firebrat embryo labelled using TUNEL (magenta) and Syncrip (white) antibodies. Syncrip was used here as a proxy for Neuroglian staining to reveal lineages. Scale bar, 50 µm. (**D**) Dying cells (arrowheads) are located close to NBs (*). Scale bar, 10 µm. (**E**) Dying cells (magenta, TUNEL) are located in the cortex of the VNC, where neurogenesis takes place, and not in the neuropil (green, Phalloidin) in a firebrat embryo. Scale bar, 50 µm.

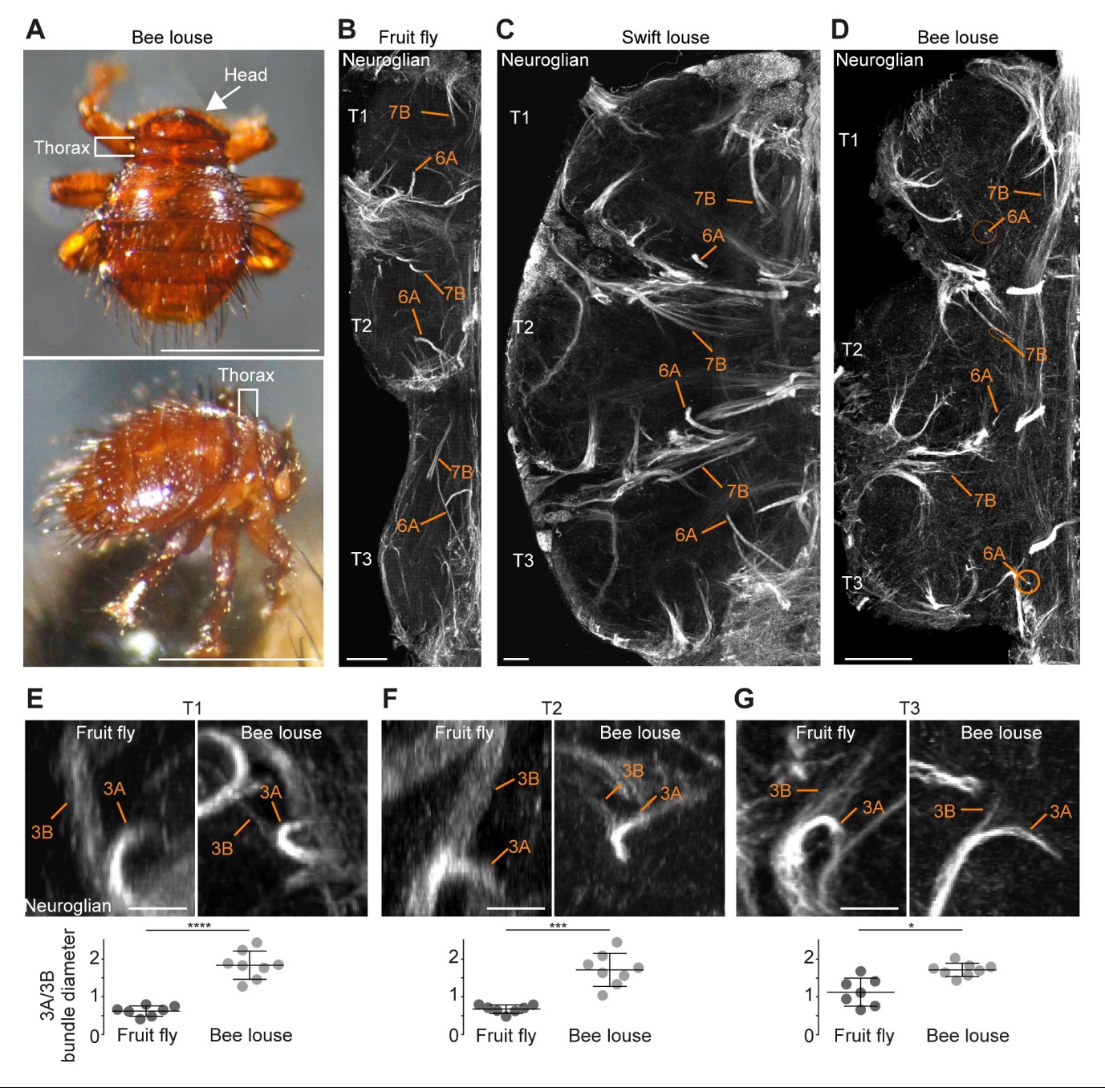

**Figure 9.** Neuronal cell death sculpts the wing circuitry of wingless dipterans. (**A**) Dorsal (top) and side view (bottom) of an adult bee lousefly with a reduced thorax lacking wings and halteres. Scale bar, 1 mm. (**B–D**) Hemilineage VNC fibre tracts labelled with Neuroglian in a fruit fly (**B**), a swift lousefly (**C**) and a bee lousefly (**D**). Shown are hemilineages 6A and 7B which are reduced in the bee lousefly in all three thoracic segments (T1, T2, T3). Scale bars, 50 µm. (**E–G**) 3A and 3B hemilineage fibre tracts labelled with Neuroglian in a fruit fly (top left) and a bee lousefly (top right). Shown are maximum intensity projections, chosen to best display hemilineages, from cross-section T1 (**E**), T2 (**F**) and frontal perspectives T3 (**G**). Quantifications of 3A/3B hemilineage bundle diameter ratios in fruit flies and bee louseflies are given below (****p < 0.0001 in T1, independent samples t-test, ***p = 0.0001 for T2, Welch's t-test, *p = 0.0044, Welch's t-test. n = 7 fruit flies each, n = 8 bee louseflies each). Bars represent mean ± standard deviation.

The online version of this article includes the following source data for figure 9:

**Source data 1.** Quantifications of 3A/3B hemilineage bundle diameter ratios in fruit flies and bee louseflies.

*Kuwada and Goodman, 1985*), although this exact same mechanism has not been demonstrated in *Drosophila*.

The spatio-temporal pattern of death revealed with SR4VH shows us that understanding the molecular control of hemilineage-based death is the key question going forward and is likely to provide insight into how networks evolve.

## Blocking death results in functional neurons that integrate into adult networks

Here we blocked PCD within the MNB lineage (called lineage 0 in the postembryonic literature) and found that 'doomed' neurons become octopaminergic, generate arborisations and target the tectulum neuropil (*Court et al., 2020*). Previous work has shown that octopamine can induce and maintain walking in locusts (*Sombati and Hoyle, 1984*) and decapitated fruit flies (*Yellman et al., 1997*). Consistently, we find that our 'undead' hemilineage 0B cells can induce walking when activated thermogenetically in headless flies and show calcium activity during naturalistic bouts of locomotion. Thus, by blocking death, we have 'resurrected' functional neurons that are able to integrate into thoracic motor networks. Although this 'dialling up' of cell numbers, in our system, is artificial, it reveals how doomed neurons possess cryptic cellular phenotypes that can emerge when death is blocked, advocating for the evolvability of such a hemilineage-based system.

While these undead hemilineage 0B octopaminergic neurons share many conserved features with wild-type cells, the variations we see in their morphology could act as a substrate for evolutionary change. A recent study has linked structural changes in the VNC with changes in behaviour between strains of *Drosophila melanogaster* (*Mellert et al., 2016*). Mellert et al., show that hemilineage 12A in the mesothorax has variable bundle morphologies and that these correlate well with the time of flight initiation. Flight is used as an escape response and can be instrumental for predator evasion, one of the major evolutionary forces which have selected for flight in insects in the first place (*Dudley, 2002*). With this in mind, it seems plausible that changes in either neuron number and/or innovations in 'undead' neuron structure could affect adult behaviour and be ultimately adaptive. Recently it has been shown that undead sensory neurons that are functional, integrate and appear to be tuned to specific odours (*Prieto-Godino et al., 2020*). Importantly, Prieto-Godino et al. also show that there is cell number variation in this neuronal population across drosophilids and that blocking PCD in *melanogaster* results in the survival of mosquito-like $CO_2$-sensing neurons in the maxillary palps. This suggests that both the central and peripheral nervous system may use similar modes of early PCD to sculpt circuits during the evolution of true flies.

## Cell death during neurogenesis is widespread across insects

Our data show that PCD is extensive and widespread during neurogenesis in the CNS of insects, from the primitive firebrats *Thermobia domestica*, to true flies *Drosophila melanogaster* and the swift lousefly *Crataerina pallida.*

We wondered whether alterations in cell death may have contributed to adaptations in the VNC of flightless dipterans and found that a greater extent of PCD in the MNB lineage may be responsible for abolishing the segment-specific difference in octopaminergic cell numbers in the swift lousefly *Crataerina pallida* – which has lost flight and gained adaptations to a parasitic lifestyle (*Bequaert, 1952*; *Hagan, 1951*; *Hutson, 1984*; *Lehane, 2008*; *Petersen et al., 2018*; *Walker and Rotherham, 2010b*). As octopaminergic neurons are involved in flight-related behaviours (*Brembs et al., 2007*; *Duch and Pflüger, 1999*; *Roeder, 2005*), we suggest that PCD has been co-opted in this lineage during the evolution of flightlessness in the swift louse. We believe that this PCD takes place early, in newly born neurons, as we have seen dying cells close to the MNB in all of our 20 pupae, from Day 4 to Day 23. Therefore, the decrease in octopaminergic cell number in the mesothoracic segment of swift louseflies is likely the result of increased hemilineage-specific PCD during evolution. Alternatively, an increase in the number of neurons in the non-flying prothorax and metathorax could lead to a uniform population size across thoracic segments in swift louseflies. This could be achieved either by additional MNB divisions, or by reduced PCD. If changes in MNB proliferation play a role remains to be determined.

## Midline neurons show hemilineage-based variations in different species

The midline neurons within the VNC of insects have long been a source of interest because they are homologous across species, yet show a diversity in cell numbers correlated with body form and function (*Lacin et al., 2019*; *Stevenson and Spörhase-Eichmann, 1995*; *Witten and Truman, 1998*). For example, flying insects have greater numbers of midline octopaminergic neurons within segments that control wings (*Campbell et al., 1995*; *Eckert et al., 1992*; *Jia and Siegler, 2002*; *Konings et al., 1988*; *Monastirioti et al., 1995*; *Schlurmann and Hausen, 2003*; *Siegler and Pankhaniya, 1997*; *Siegler et al., 2001*; *Siegler et al., 1991*; *Spörhase-Eichmann et al., 1992*; *Stevenson et al., 1992*; *Thompson and Siegler, 1993*), while grasshoppers have more GABAergic neurons in the metathoracic/abdominal ganglia, which receives auditory input (*Witten and Truman, 1998*; *Thompson and Siegler, 1991*). These two midline neuronal populations are derived from the same NB, the MNB, which buds off multiple GMCs, each dividing once to generate a GABAergic (A cell) and an octopaminergic (B cell) neuron. Both A and B neurons are generated in equal numbers but, in all cases, the numbers of GABAergic ('hemilineage A') and octopaminergic ('hemilineage B') neurons within one segment are never the same. The greater numbers of GABAergic cells within each segment has been shown in grasshoppers (*Jia and Siegler, 2002*) and fruit flies (*Truman et al., 2010*) to be the result of removal by PCD of large numbers of cells from hemilineage B.

As suggested by our previous work, the 'hemilineage' emerges as a discrete developmental unit that shows common features of gene expression and function (*Harris et al., 2015*; *Lacin et al., 2019*; *Lacin and Truman, 2016*; *Shepherd et al., 2019*; *Truman et al., 2010*). Therefore, we refer here to the PCD found in the MNB lineage of grasshoppers as 'hemilineage-specific PCD'. Following our observations of PCD during development in other insects, together with a vast body of knowledge on homology in insect nervous systems (*Kutsch and Breidbach, 1994*; *Thomas et al., 1984*), we suggest that variations in neural circuits between species is very likely set up by modifying hemilineages, with PCD playing a major role.

## Hemilineage-specific reductions in fibre diameter in the bee lousefly

In the wingless bee lousefly *Braula coeca*, we found clear reductions in the thickness of fibre tracts in several hemilineage bundles which in *Drosophila* are associated with flight-related behaviours (*Harris et al., 2015*). It remains for us to determine if early PCD takes place in these specific bee lice lineages during development and causes the reduction in bundle diameter that we see in flight hemilineages. As with most parasitic insects, bee and swift louseflies are impossible to maintain in the laboratory in the absence of their hosts and procuring them is a challenge (e.g. bee louseflies are now only found on two islands in the UK, while collecting swift louseflies is restricted to the summer months due to *Apus apus* migrations). Nonetheless, our observations that PCD is widespread across insects complements our findings in bee louseflies, strongly suggesting that an extensive PCD in flight hemilineages may have accompanied the loss of flight during evolution. Interestingly, the reduction in fibre diameter we see in bee louseflies was not evident in swift louseflies. This difference is likely due to the more significant changes to body plan in bee louseflies, i.e. a complete loss of the flight apparatus during evolution. Swift louseflies however still maintain vestigial wings and halteres (*Walker and Rotherham, 2010b*), whereas bee louseflies have a severely reduced thorax, completely lacking wings, halteres and flight muscles (*McAlister, 2018*).

## Conclusions

Here we have shown that undead neurons elaborate complex arborisations, express distinct transmitter identities and function. We find that 'early' PCD is widespread during the development of the CNS of insects from the primitive firebrats, to most derived true flies. Early cell death appears to be a specific subtype of PCD present across animals. Understanding how early PCD is specified across species should help us elucidate how nervous systems are built and evolve. Our exploration of homologous lineages in flightless dipterans shows that changes in body plan may accompany changes in the extent and pattern of PCD. As the evolutionary changes seen in neural networks ultimately result from heritable differences in developmental processes, our future endeavours will be directed towards elucidating how genetic programs are deployed to establish the pattern of PCD. The cellular leitmotif of hemilineage-based cell death, we present here, provides us with something tangible that we can search for. Thus, we suggest that viewing the evolution of insect nervous

systems through the lens of the 'hemilineage' will be critical for understanding how development brings about adaptive changes in neural network motifs.

# Materials and methods

## Key resources table

| Reagent type (species) or resource | Designation | Source or reference | Identifiers | Additional information |
|---|---|---|---|---|
| Gene (*Drosophila melanogaster*) | worniu | FlyBase | FLYB:FBgn0001983 | |
| Gene (*Drosophila melanogaster*) | TDC2 | FlyBase | FLYB:FBgn0050446 | |
| Gene (*Drosophila melanogaster*) | VGlut | FlyBase | FLYB:FBgn0031424 | |
| Gene (*Drosophila melanogaster*) | hid | FlyBase | FLYB:FBgn0003997 | |
| Gene (*Drosophila melanogaster*) | grim | FlyBase | FLYB:FBgn0015946 | |
| Gene (*Drosophila melanogaster*) | rpr | FlyBase | FLYB:FBgn0011706 | |
| Gene (*Drosophila melanogaster*) | skl | FlyBase | FLYB:FBgn0036786 | |
| Gene (*Drosophila melanogaster*) | Dronc | FlyBase | FLYB:FBgn0026404 | |
| Genetic reagent (*Drosophila melanogaster*) | Worniu-GAL4 | Bloomington *Drosophila* Stock Center | BDSC:56553; FLYB:FBtp0021524; RRID:BDSC_56553 | FlyBase symbol: P{wor.GAL4.A} |
| Genetic reagent (*Drosophila melanogaster*) | Tdc2-GAL4 | Bloomington *Drosophila* Stock Center | BDSC:9313; FLYB:FBtp0127561; RRID:BDSC_9313 | FlyBase symbol: P{Tdc2-GAL4.C} |
| Genetic reagent (*Drosophila melanogaster*) | OK371-GAL4 | Bloomington *Drosophila* Stock Center | BDSC: 26160; FLYB:FBti0076967; RRID:BDSC_26160 | FlyBase symbol: Dmel\P{GawB}VGlut$^{OK371}$ |
| Genetic reagent (*Drosophila melanogaster*) | UAS-SR4VH | This paper | | Fly line maintained in DW Williams lab; See Materials and methods, section 'Contstruction of *UAS-SR4VH*' |
| Genetic reagent (*Drosophila melanogaster*) | UAS-CD8::GFP | Bloomington *Drosophila* Stock Center | BDSC:5137; FLYB:FBtp0002652; RRID:BDSC_5137 | FlyBase symbol: P{UAS-mCD8::GFP.L} |
| Genetic reagent (*Drosophila melanogaster*) | UAS-tdTomato-p2A-GCaMP6s | *Chen et al., 2018* | | Gift from MH Dickinson |
| Genetic reagent (*Drosophila melanogaster*) | H99 | Bloomington *Drosophila* Stock Center | BDSC:1576; FLYB:FBab0022359; RRID:BDSC_1576 | FlyBase symbol: Df(3L)H99 |
| Genetic reagent (*Drosophila melanogaster*) | XR38 | *Peterson et al., 2002* | FLYB:FBab0027961 | FlyBase symbol: Df(3L)XR38 |
| Genetic reagent (*Drosophila melanogaster*) | dronc$^{\Delta A8}$ | *Kondo et al., 2006* | FLYB:FBal0244156 | FlyBase symbol: Dronc$^{\Delta A8}$ |
| Biological sample (*Crataerina pallida*) | Swift lousefly | Collected from Cambridgeshire and Suffolk, UK | | Whole CNS or just VNC freshly dissected from *Crataerina pallida* |

*Continued on next page*

*Continued*

| Reagent type (species) or resource | Designation | Source or reference | Identifiers | Additional information |
|---|---|---|---|---|
| Biological sample (*Braula coeca*) | Bee lousefly | A. Abrahams, Isle of Colonsay, UK | | VNC freshly dissected from *Braula coeca* |
| Biological sample (*Thermobia domestica*) | Firebrat | Buzzard Reptile and Aquatics (buzzardreptile.co.uk) | | VNC freshly dissected from *Thermobia domestica* |
| Antibody | anti-GFP (Chicken polyclonal) | Abcam | Cat# ab13970, RRID:AB_300798 | IF(1:500) |
| Antibody | anti-Neuroglian (Mouse monoclonal) | DSHB | Cat# BP 104, RRID:AB_528402 | IF(1:50) |
| Antibody | anti-Cleaved *Drosophila* Dcp-1 (Rabbit polyclonal) | Cell Signaling Technology | Cat# 9578, RRID:AB_2721060 | IF(1:100) |
| Antibody | anti-Syncrip (Guinea pig polyclonal) | Gift from I. Davis | | IF(1:100) |
| Antibody | anti-Engrailed/Invected (Mouse monoclonal) | DSHB | Cat# 4D9, RRID:AB_528224 | IF(1:2) |
| Antibody | anti-DVGLUT C-terminus (Rabbit polyclonal) | Gift from H. Aberle | Cat# AB-DVGLUT-C, RRID:AB_2490071 | IF(1:5000) |
| Antibody | anti-tyramine β-hydroxylase (Rat monoclonal) | Gift from M. Monastirioti | Cat# Tyramine beta-hydroxylase (TBH), RRID:AB_2315520 | IF(1:50) |
| Antibody | anti-vestigial (Rabbit polyclonal) | Gift from S. Carroll and K. Gruss | | IF(1:400) |
| Antibody | anti-GABA (Rabbit polyclonal) | ImmunoStar | Cat# 20094, RRID:AB_572234 | IF(1:100) |
| Recombinant DNA reagent | pUAST (plasmid) | **Brand and Perrimon, 1993** | | Insect expression, *Drosophila*; See Materials and methods, section 'Contstruction of *UAS-SR4VH*' |
| Commercial assay or kit | Click-iT EdU Cell Proliferation Kit for Imaging | *life* technologies | Cat# C10337 | |
| Commercial assay or kit | Click-iT Plus TUNEL Assay for In Situ Apoptosis Detection | *life* technologies | Cat# C10618 | |
| Software, algorithm | Python Programming Language | Python Programming Language | RRID:SCR_008394 | |
| Software, algorithm | MATLAB | MathWorks | RRID:SCR_001622 | |
| Software, algorithm | SPSS | IBM | RRID:SCR_002865 | |
| Software, algorithm | Fiji | Fiji | RRID:SCR_002285 | |
| Other | DAPI | Sigma-Aldrich | Cat# D9542 | (1 μg/mL) |
| Other | Phalloidin-488 | *life* technologies | Cat# A12379 | (1:100) |

## Animals

We used the following *Drosophila melanogaster* stocks: *Worniu-GAL4; Dr/TM3, Ubx-LacZ, Sb* (BDSC_56553), *TDC2-GAL4* (BDSC_9313), *OK371-GAL4* (BDSC_26160), *UAS-SR4VH* (described here), *UAS-CD8::GFP* (BDSC_5137), *UAS-tdTomato-p2A-GCaMP6s* (**Chen et al., 2018**) (kind gift from M. Dickinson), *H99/TM3, Sb* (BDSC_1576), *XR38/TM3* (**Peterson et al., 2002**), Sb, *If/CyO;*

$dronc^{\Delta A8}$, *FRT2A/TM6β*, *Tb*, *Hu* (**Kondo et al., 2006**) and *hs-flp;; TubP-GAL80, FRT2A/TM3, Sb* (**Truman et al., 2010**).

Firebrat adults of *Thermobia domestica* were obtained from Buzzard Reptile and Aquatics (buzzardreptile.co.uk) and reared on a diet of fish flakes and wholemeal bran at 40°C in darkness inside a humid plastic container. A staging series was calculated by time to hatching.

*Crataerina pallida* swift lousefly adults were collected from swift (*Apus apus*) nesting boxes fitted behind the louvres of belfry windows from churches in Cambridgeshire and Suffolk (UK) with the help of local conservationists Simon Evans, Richard Newell and Bill Murrells. Swift louseflies were kept at 20°C on a 12 hr dark:12 hr light cycle until dissected. Pregnant females, recognised by their enlarged and translucent abdomen through which larvae or prepupae could be detected, were kept separately and checked daily for pupa ejection. The day in which a pupa was laid was defined as 'Day 0' of external development (outside the mother's abdomen).

*Braula coeca* bee lousefly adults were obtained from a black bee (*Apis mellifera mellifera*) colony on the Isle of Colonsay, UK (kind gift from A. Abrahams). Bee louseflies were shipped by post in small cages containing worker bees feeding on bee fondant. The black bees and bee louseflies were anaesthetised by placing the cage on a $CO_2$ pad and the bee louseflies were removed for dissection.

## Construction of *UAS-SR4VH*

SR4VH was constructed by standard molecular biology procedures. It comprises the myristylation signal of *Drosophila* Src64B (amino acids 1–95), a monomeric red fluorescent protein mRFP1 (**Campbell et al., 2002**), a linker that contains four DEVD sites, a yellow florescent protein Venus (**Nagai et al., 2002**), and a nuclear localisation signal of *Drosophila* histone H2B (amino acids 1–51). While the design is similar to the previously reported caspase probe Apoliner (**Bardet et al., 2008**), the Src64B myristylation signal and the H2B NLS offers better membrane and nuclear localisation, respectively, and four DEVD sites are expected to provide higher sensitivity. The probe was cloned in pUAST (**Brand and Perrimon, 1993**) and introduced into the *Drosophila* genome by P element-mediated transformation.

## Immunohistochemistry and chemical staining

*Drosophila* larvae were dissected in PBS without anaesthesia. Firebrat embryos were removed from their chorion and dissected using minuten pins. *Drosophila*, swift lousefly and bee lousefly adults were anaesthetised on ice, briefly submerged in absolute ethanol and dissected in PBS. Swift louse-fly pupae were immobilised on double sided sticky tape, removed from their pupal case using forceps and dissected in PBS without anaesthesia. Samples were fixed in 3.6% paraformaldehyde in PBS for 30 min (larvae and pupae) or 1 hr (adults), washed 3 times in 0.3% PBST (0.3% Triton-X100 in PBS, Sigma-Aldrich), blocked in 5% goat serum (Sigma-Aldrich) in PBST for 1 hr and incubated with primary antibodies in block for 1–3 days at 4°C (*Drosophila*, bee louseflies, swift lousefly pupae), room temperature (firebrats) or 37°C to increase antibody penetration (swift lousefly adults; block supplemented with 0.02% $NaN_3$ to prevent microbial growth). Samples were then washed four times throughout the day in PBST and incubated with secondary antibodies in block for a further 1–3 days, followed by final washes in PBST and PBS. Brains and VNCs were mounted on poly-L-lysine-coated coverslip, dehydrated in increasing serial concentrations of ethanol (15%, 30%, 70%, 80%, 90% and twice in 100%) for 5 min each, dipped once in xylene, then incubated twice for 5 min in fresh xylene. A droplet of DePeX (EMS) was added on top of the mounted sample and the coverslip was placed face-down on a glass slide.

We used the following primary antibodies: chicken anti-GFP (1:500; ab13970, Abcam), mouse anti-Neuroglian (1:50; BP 104, Developmental Studies Hybridoma Bank), rabbit anti-cleaved *Drosophila* Dcp1 (1:100; 9578, Cell Signaling), guinea pig anti-Syncrip (1:100; kind gift from I. Davis; to label NBs and early progeny in lineages - JW Truman, personal communication, January 2019), mouse anti-Engrailed/Invected (1:2; 4D9, Developmental Studies Hybridoma Bank), rabbit anti-DVGLUT C-terminus (**Mahr and Aberle, 2006**) (1:5000; AB_2490071, kind gift from H. Aberle), rat anti-tyramine β-hydroxylase (**Monastirioti et al., 1996**) (1:50; AB_2315520, kind gift from M. Monastirioti), rabbit anti-vestigial (1:400; kind gift from Sean Carroll and Kirsten Gruss) and rabbit anti-GABA (1:100; 20094, ImunoStar).

Secondary antibodies were Alexa Fluor 488-conjugated goat anti-chicken (1:500; A11039, Invitrogen, Thermo Fisher Scientific), Alexa Fluor 488-conjugated goat anti-rabbit (1:500; A11070, Invitrogen, Thermo Fisher Scientific), Cy3-conjugated donkey anti-rabbit (1:500; 711-006-152, Jackson ImmunoResearch), Cy5-conjugated donkey anti-mouse (1:500; 715-006-151, Jackson ImmunoResearch), Alexa Fluor 488-conjugated donkey anti-rat cross-adsorbed against mouse (1:100; 712-545-153, Jackson ImmunoResearch), Alexa Fluor 488-conjugated donkey anti-guinea pig (1:500; A11073, Invitrogen, Thermo Fisher Scientific).

In firebrat embryos we detected dying cells using the Click-iT Plus TUNEL assay kit (C10618, *life* technologies). To stain cell nuclei and neuropil, firebrat samples were incubated with DAPI (1 µg/mL; D9542, Sigma-Aldrich) and Phalloidin-488 (1:100, A12379, *life* technologies) in PBST for 30 min at room temperature. Incubations were carried out following secondary antibody treatment. Samples were then washed in PBST and PBS, and mounted.

## EdU treatment

To label proliferating cells and their progeny we used the Click-iT EdU imaging Kit (C10337, *life* technologies). Freshly dissected nervous systems from swift lousefly pupae were incubated in EdU 1:1000 in PBS at room temperature for 1–3 hr on a shaker, rinsed with PBS and fixed in cold buffered formalin 3.6% in PBS for 30 min. Samples were then stained using the immunohistochemistry protocol described above. The colour reaction for EdU was carried out as instructed by the vendor after the secondary antibodies were washed out.

## Generation of undead neuron MARCM clones

To induce mitotic clones of undead neurons, rescued from PCD, we used the mosaic analysis with a repressible cell marker technique (*Lee and Luo, 1999*). 0–4 hr first instar larvae resulting from crossing females of the genotype *hs-flp;; TubP-GAL80, FRT2A/TM3, Sb* with; *TDC2-GAL4, UAS-CD8:: GFP, UAS-TrpA1; dronc^{ΔA8}, FRT2A/TM6β, Tb, Hu* males were heat-shocked at 37°C in a plastic food vial placed in a water bath for either 1 hr or 45 min, followed by 45 min at room temperature and a second incubation period at 37°C for 30 min. After heat-shock, larvae were immediately returned to 23°C or 25°C. Cell death was blocked in clones homozygous for the loss-of-function allele of the initiator caspase Dronc. Because we used the octopaminergic driver line *TDC2-GAL4* to induce the expression of CD8::GFP and TrpA1, we were able to visualise and thermogenetically activate only postembryonic neurons of hemilineage 0B. A small number of wild-type octopaminergic neurons are born during postembryonic neurogenesis (one in T1 and T3, 4–5 in T2, see *Figure 3—figure supplement 1*). To ensure the characterisation of undead neurons only, MARCM clones including a bilaterally symmetrical primary neurite were excluded from analysis.

## Thermogenetic activation and video recordings

Prior to recordings, 2–6 day-old males of *Drosophila melanogaster* (*hs-flp/+; TDC2-GAL4, UAS-CD8::GFP, UAS-dTRPA1/+; dronc^{ΔA8}, FRT2A/TubP-GAL80, FRT2A*) reared at 23°C or 25°C in a 12 hr:12 hr light:dark cycle were anaesthetised on ice and decapitated using a pair of micro spring scissors in under 3 min. We used males as we found they are more responsive to octopamine release by thermogenetic activation than females (data not shown). The headless flies were brushed back into a food vial placed on its side and left to recover for at least 1 hr. To generate the heat ramp required to thermogenetically activate undead neurons, we used a 12V thermoelectric Peltier plate (model: TEC1-12706, size: 40 mm x 40 mm x 3.6 mm) connected to a DC power supply (HY3005D, Rapid Electronics) set at a constant current of 0.46A, with a variable voltage, calibrated using an infrared laser thermometer (N92FX, Maplin). These settings generated a temperature ramp which lasted 70 s from 22°C to 34°C. Videos were recorded at 25 fps using a Sony NEX-5N digital camera (kindly provided by Ian Wynne) mounted to a stereo microscope. A piece of graph paper was used for spatial calibration. To match the presence of undead neurons with behaviour, each decapitated fly used for thermogenetic activation was indexed and prepared for dissection and immunostaining.

## Two-photon calcium imaging in behaving intact flies

The method for in vivo two-photon imaging of the VNC in behaving adult *Drosophila* is described in *Chen et al., 2018*. Briefly, flies were anaesthetised through cooling and then mounted onto custom

imaging stages. The dorsal thoracic cuticle was removed and indirect flight muscles were left to degrade over the course of 1 hr. Subsequently, the proventriculus and salivary glands were resected to gain optical access to the VNC.

Horizontal sections of the T1 leg ganglion were imaged using galvo-galvo scanning. For control animals, the bifurcation point of TDC-positive neurites were imaged to circumvent ROI disappearances caused by movement. For animals harbouring undead *TDC2-GAL4*-positive neurons, the thickest branch of the axonal bifurcation was chosen because they were most likely to contain undead neurites. Image dimensions ranged between $512 \times 512$ and $320 \times 320$, resulting in 1.6 to 3.4 fps data acquisition. Imaging areas ranged between $92 \times 92$ µm and $149 \times 149$ µm. Laser power was held at ~8 mW.

## Data analysis for 2-photon imaging in behaving *Drosophila*

Python scripts (modified from *Chen et al., 2018*) were used to extract ROI fluorescence traces and to compute spherical treadmill ball rotations. Walking epochs were determined by placing a threshold on ball rotations, which were first converted into anterior-posterior ($v_{forward}$) and medial-lateral ($v_{side}$) speeds (one rot s$^{-1}$ = 31.42 mm s$^{-1}$) and into degrees s$^{-1}$ (1 rot s$^{-1}$ = 360° s$^{-1}$) for yaw ($v_{rotation}$) movements. Thresholds were 0.12 mm, 0.12 mm and 5°, respectively. Periods below these thresholds were considered 'resting' while other periods were considered 'walking'. Fluorescence traces for epochs with the same behaviour were aligned by start point to compute average %ΔR/R traces for specific actions.

To calculate fluorescence traces for small subregions-of-interest across neuritic bundles containing both undead and wild-type neurites, images were registered using an optic flow method described in *Chen et al., 2018*. This registration served to minimise motion artefacts. Analysis was limited to a period with no warping artefacts and no ROI disappearance. Subregions were manually selected as small circular ROIs across the neuritic bundle of the registered image. Fluorescence values were then computed from each sub-ROI.

## Confocal imaging and image processing

Images were acquired using a Zeiss LSM 510 or a Zeiss LSM 800 confocal microscope at a magnification of 20x or 40x with optical sections taken at 1 µm intervals. The resulting images were examined and processed using Fiji (https://imagej.net/Fiji). Some images were manually cropped using the Freehand Selection tool to remove debris or to cut out neuronal lineages in *Worniu-GAL4, UAS-SR4VH* samples.

## Fluorescence intensity plots

To generate fluorescence intensity along Line plots, we used the Plot Profile tool in Fiji to extract raw fluorescence intensity values for the RFP and Venus channels. The values were imported into MATLAB (R2018a, MathWorks) and normalised by dividing all fluorescence intensity values to the maximum value encountered along each Line. In this manner, all fluorescence intensity along Line plots have a common scale from 0 to 1, with one being the highest value encountered along that Line.

## Analysis of thermogenetic activation

Decapitated flies were considered to be walking if they covered a distance greater than one body length and moved their legs in a coordinated sequence from T3 to T2 to T1 at least once on each side (*Harris et al., 2015*). Forward, backward and sideway movements were all interpreted as walking when both aforementioned conditions were respected. To generate fly body traces video recordings were imported in MATLAB (R2018a, MathWorks) and the centroid of the decapitated fly (located on the scutellum) was extracted from each frame using a custom-written script which can be found at github.com/snznpp/undead-walking (*Pop, 2020*; copy archived at https://github.com/elifesciences-publications/undead-walking). Each frame was converted into a greyscale image, its contrast enhanced using contrast-limited adaptive histogram equalisation, filtered using a Gaussian smoothing kernel with a standard deviation of 4, binarised using a custom threshold and the geometric centre of the fly body automatically extracted and stored in an array. To confirm that the

centroid detection was accurate, a red dot with the centroid coordinates was superimposed onto each frame of the original recording and the annotated movie was saved for manual inspection.

## Quantification of 3A/3B bundle diameters

For calculating 3A/3B hemilineage bundle diameter ratios in fruit flies and bee louseflies, we generated transverse rendered maximum intensity projections of inverted greyscale confocal stacks for the pro- and mesothorax (T1 and T2) and frontal projections for the metathorax (T3). Optical sections were selected to include the common lineage bundle and the individual hemilineage bundles after their split. Diameter measurements were taken at the widest point within 5 μm of the bundle split using the Straight Line tool in Fiji and ratios were calculated by dividing the diameter of hemilineage 3A to that of 3B.

## Statistical analysis

For comparing neuron numbers, 3A/3B bundle diameter and T2/T1 number of neurons, data were tested for normal distribution using the Kolmogorov-Smirnov test and visualisation of Normal Q-Q plots. Differences between groups were analysed using either the independent samples t-test for normally distributed data, Welch's test if data failed to meet the homogeneity of variances assumption or Mann-Whitney t-tests if data failed to meet the normality and homogeneity of variances assumptions of the independent samples t-test.

For comparing the number of flies which walked in each experimental group, we performed a Pearson chi-squared test and interpreted the resulting exact significance if the minimum expected count was greater than 5, or the Fisher's Exact Test 2-sided significance if the minimum expected count was lower than five in at least one cell of the contingency table. To correct for multiple comparisons we performed a Bonferroni correction (i.e. p values were multiplied by 6, the total number of pairwise tests).

All statistical tests were performed in SPSS Statistics 23 (IBM) with an $\alpha$ set at 0.05. In all figures, bars represent means $\pm$ standard deviation; $*p < 0.05$, $***p < 0.001$, $^{ns}p$ = not significant.

# Acknowledgements

We would like to thank Richard Benton and Lucia Prieto-Godino for discussions and sharing data. We thank Kristin White and Bloomington *Drosophila* Stock Center (NIH P40OD018537) for sharing flies; Maria Monastirioti, Hermann Aberle, Ilan Davis, Kirsten Gruss, Sean Carroll and Developmental Studies Hybridoma Bank (NICHD of the NIH, University of Iowa) for antibodies. We are indebted to Simon Evans, Richard Newell and Bill Murrells for their kind help in collecting swift lice and Andrew Abrams for sending us bee lice. We are grateful to Andrew M Dacks and Hans-Joachim Pflüger for providing unpublished data on *Manduca* octopaminergic neurons. We would like to thank Ian Wynne for his camera. We also thank Matthias Landgraf, David Shepherd, Jon Clarke and Sanjay Sane for reading the manuscript. Funding: Williams: BBSRC BB/P025552/1 and BB/L022672/1. Ramdya: SNSF Project Grant: 175667; Eccellenza Grant: 181239; R'Equip Grant: 177102.

# Additional information

## Funding

| Funder | Grant reference number | Author |
| --- | --- | --- |
| Biotechnology and Biological Sciences Research Council | BB/P025552/1 | Darren W Williams |
| Biotechnology and Biological Sciences Research Council | BB/L022672/1 | Darren W Williams |
| Schweizerischer Nationalfonds zur Förderung der Wissenschaftlichen Forschung | 175667 | Pavan Ramdya |
| Schweizerischer Nationalfonds zur Förderung der Wissenschaftlichen Forschung | Eccellenza 181239 | Pavan Ramdya |

| Schweizerischer Nationalfonds zur Förderung der Wissenschaftlichen Forschung | R'Equip 177102 | Pavan Ramdya |

The funders had no role in study design, data collection and interpretation, or the decision to submit the work for publication.

## Author contributions
Sinziana Pop, Conceptualization, Resources, Data curation, Software, Validation, Investigation, Visualization, Methodology, Writing - original draft, Writing - review and editing; Chin-Lin Chen, Connor J Sproston, Resources, Investigation, Visualization, Methodology, Writing - review and editing; Shu Kondo, Resources, Validation, Methodology, Writing - review and editing; Pavan Ramdya, Resources, Supervision, Funding acquisition, Visualization, Methodology, Writing - review and editing; Darren W Williams, Conceptualization, Resources, Data curation, Supervision, Funding acquisition, Investigation, Visualization, Methodology, Writing - original draft, Project administration, Writing - review and editing

## Author ORCIDs
Sinziana Pop (iD) https://orcid.org/0000-0002-8811-8307
Chin-Lin Chen (iD) https://orcid.org/0000-0002-4968-4920
Connor J Sproston (iD) https://orcid.org/0000-0003-2491-0589
Shu Kondo (iD) https://orcid.org/0000-0002-4625-8379
Pavan Ramdya (iD) https://orcid.org/0000-0001-5425-4610
Darren W Williams (iD) https://orcid.org/0000-0001-5917-4935

## Decision letter and Author response
Decision letter https://doi.org/10.7554/eLife.59566.sa1
Author response https://doi.org/10.7554/eLife.59566.sa2

## Additional files
### Supplementary files
• Transparent reporting form

## Data availability
All data generated or analysed during this study are included in the manuscript and supporting figures.

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
