## [Decision Letter]

**Acceptance summary:**

This study by Pop et al. is a bold approach to understanding the mechanisms that shape neural circuitry during evolution. It takes advantage of the idea that the bauplan of insect nervous systems is highly conserved and asks how programmed cell death (PCD) might be used in different species to construct their neural circuitry.

**Decision letter after peer review:**

Thank you for submitting your article "Extensive and diverse patterns of neuronal cell death sculpt neural networks in insects" for consideration by *eLife*. Your article has been reviewed by three peer reviewers, and the evaluation has been overseen by K VijayRaghavan as the Senior and Reviewing Editor. The reviewers have opted to remain anonymous.

The reviewers have discussed the reviews with one another and the Reviewing Editor has drafted this decision to help you prepare a revised submission.

Summary:

This study by Pop et al. is a bold approach to understanding the mechanisms that shape neural circuitry during evolution. It takes advantage of the idea that the bauplan of insect nervous systems is highly conserved and asks how programmed cell death (PCD) might be used in different species to construct their neural circuitry. First, the authors present comparative data illustrating high levels of programmed cell death in neurons that would likely have been associated with flight circuitry in insects that have lost flight. Then, by exploiting a new tool that they have developed to monitor cell death, they illustrate that in *Drosophila* programmed cell death occurs soon after neurons are born. By blocking cell death, they demonstrate that these "undead" neurons can be integrated into normal neural circuits and can drive behavior. The data are of high quality and well presented, and the manuscript is well written.

In our consultations, we all felt that the paper is written almost backwards from how we may have organized the data. The *Drosophila* results provide a compelling case that neurons are eliminated early after birth and that blocking their death can lead them to be integrated into functional circuits. The comparative work provides examples of how these results from *Drosophila* may generalize and may contribute to adaptation, should be presented later. This ordering would also have the salubrious effect of not encouraging readers to think that the authors should have done the manipulative experiments in the non-model species. The way it is structured now, the thought immediately occurs, "Why don't they block programmed cell death in the species that have lost most of the neurons?" Then one realizes that this experiment is probably technically impossible at the moment, given the challenges of rearing these species. By rearranging the results, the readers would not jump to this idea as an "obvious" experiment.

Essential revisions:

1) There is no direct quantification of PCD between the different species, instead there are correlative measures. Therefore, the language surrounding the implications of Figures 2 and 3 needs to be tempered in regard to the stated goal in the subsection “Early mode of neuronal cell death is found in many different insects” (determining if the loss of flight resulted in increased PCD). This goal is difficult to meet without directly measuring differences in PCD and the comparative approach can only show correlation, not causation. In Figure 2 they use a ratio of the number of OA neurons present in the pro- vs. mesothoracic neuromeres as a correlate of PCD as previous work has shown that PCD eliminates OAergic neurons in some species. The assumption made here is that if there is a lower ratio of OA neurons in the pro vs. mesothoracic neuromeres, then there must have been more PCD in the mesothoracic segments for the flightless flies. However, could it not be just as likely that there is increased PCD in non-winged neuromeres in flying insects? There is a good deal of interspecies variability in the number of OA neurons within each neuromere, so it's hard to attribute loss of flight for differences in these ratios which is the stated in the aforementioned subsection. The same sort of concern can be raised for the data in Figure 3. The authors show a decrease in the bundle width of wing vs. leg neuropil innervating hemilineages, but again without quantification of PCD, this should be framed as correlational. In addition, there is a danger in placing too little stock in the possibility of non-flight related roles for OA neurons. Please note that we are not requesting additional experiments here.

2) The walking behaviour experiments. The Tdc2-Gal4 will be expressed in all 'undead' octopaminergic neurons (including those from the MNB). It will also be expressed in 'WT' octopaminergic neurons (that are dronc-/-). Because of this, it will not be possible for the authors to determine whether bouts of walking in the headless fly were initiated due to the activation of the 'undead' octopaminergic neurons from the MNB or from any of the other octopaminergic neurons. The authors seem to have images of each walking fly's thorax (5E). Can they examine these to determine whether all the animals that walked had 'undead' neurons from the MNB lineage?

3) The functional imaging experiments: The authors reason that undead neurons outnumber their wild-type counterparts, and therefore their Ca^2+^ signal must be from the undead neurons. One can buy that, but we wonder if, in their images, the authors also have any of the neurites that are specific to the undead cells. Ca^2+^ signals from there will be irrefutable.

4) A general comment that is also stated in the summary. We found each result in this manuscript interesting. Yet, there is a discontinuity in the way the results are stitched together. While the evolutionary comparisons are to do with flight and the lineages related to flight, the latter part of the story that deals with the 'undead' neurons relate to walking. We can't think of an appropriate fix for this. If the authors were to ignore flight, the evolutionary angle is meaningless; if they were to focus on the evolutionary angle (which is lovely), the functionality of the undead neurons must be assessed in the light of flight, not walking. It is possible that the Lineage 0B modulates both flight and walking – locomotion in general. In this case the authors could discuss upfront that walking is being used in their functional imaging experiments because it's an easier prep that will nevertheless report on connectivity of the undead neurons.

5) Do the authors have data looking at markers of PCD in the bee and swift louseflies? If so please include them. If not, given the challenges of COVID-19 and the difficulty of obtaining these animals we do not request these as new experiments.

---

## [Author Response]

In our consultations, we all felt that the paper is written almost backwards from how we may have organized the data. The Drosophila results provide a compelling case that neurons are eliminated early after birth and that blocking their death can lead them to be integrated into functional circuits. The comparative work provides examples of how these results from Drosophila may generalize and may contribute to adaptation, should be presented later. This ordering would also have the salubrious effect of not encouraging readers to think that the authors should have done the manipulative experiments in the non-model species. The way it is structured now, the thought immediately occurs, "Why don't they block programmed cell death in the species that have lost most of the neurons?" Then one realizes that this experiment is probably technically impossible at the moment, given the challenges of rearing these species. By rearranging the results, the readers would not jump to this idea as an "obvious" experiment.

Paper rearranged: > early PCD in *melanogaster*, undead neurons, comparative work.

Essential revisions:1) There is no direct quantification of PCD between the different species, instead there are correlative measures. Therefore, the language surrounding the implications of Figures 2 and 3 needs to be tempered in regard to the stated goal in the subsection “Early mode of neuronal cell death is found in many different insects” (determining if the loss of flight resulted in increased PCD). This goal is difficult to meet without directly measuring differences in PCD and the comparative approach can only show correlation, not causation. In Figure 2 they use a ratio of the number of OA neurons present in the pro- vs. mesothoracic neuromeres as a correlate of PCD as previous work has shown that PCD eliminates OAergic neurons in some species. The assumption made here is that if there is a lower ratio of OA neurons in the pro vs. mesothoracic neuromeres, then there must have been more PCD in the mesothoracic segments for the flightless flies. However, could it not be just as likely that there is increased PCD in non-winged neuromeres in flying insects? There is a good deal of interspecies variability in the number of OA neurons within each neuromere, so it's hard to attribute loss of flight for differences in these ratios which is the stated in the aforementioned subsection. The same sort of concern can be raised for the data in Figure 3. The authors show a decrease in the bundle width of wing vs. leg neuropil innervating hemilineages, but again without quantification of PCD, this should be framed as correlational. In addition, there is a danger in placing too little stock in the possibility of non-flight related roles for OA neurons. Please note that we are not requesting additional experiments here.

We changed the language around the involvement of PCD in the evolution of flightlessness from causative to correlational, see Results and Discussion. We discussed that changes in the MNB lineage in the swift lousefly may be due to changes in proliferation. We clarify that octopaminergic neurons have a role in both walking and flight, Introduction section.

2) The walking behaviour experiments. The Tdc2-Gal4 will be expressed in all 'undead' octopaminergic neurons (including those from the MNB). It will also be expressed in 'WT' octopaminergic neurons (that are dronc-/-). Because of this, it will not be possible for the authors to determine whether bouts of walking in the headless fly were initiated due to the activation of the 'undead' octopaminergic neurons from the MNB or from any of the other octopaminergic neurons. The authors seem to have images of each walking fly's thorax (5E). Can they examine these to determine whether all the animals that walked had 'undead' neurons from the MNB lineage?

Because of MARCM, the TDC2-Gal4 is only expressed in dronc+/+ neurons. To clarify, we include Figure 3—figure supplement 2 and clarify in subsection “Undead neurons are functional and integrate into motor networks”.

3) The functional imaging experiments: The authors reason that undead neurons outnumber their wild-type counterparts, and therefore their Ca^2+^ signal must be from the undead neurons. One can buy that, but we wonder if, in their images, the authors also have any of the neurites that are specific to the undead cells. Ca^2+^ signals from there will be irrefutable.

Due to the complex nature of this experiment, we used H99/XR38 where it’s impossible to separate WT from undead. We justify our choice of fly and clarify that the thicker bundle of the bifurcation must contain mostly undead neurons in the Materials and methods. Additionally, Figure 3—figure supplement 2 of MARCM undead neurons helps us make the point that all undead neurons take a turn collectively.

4) A general comment that is also stated in the summary. We found each result in this manuscript interesting. Yet, there is a discontinuity in the way the results are stitched together. While the evolutionary comparisons are to do with flight and the lineages related to flight, the latter part of the story that deals with the 'undead' neurons relate to walking. We can't think of an appropriate fix for this. If the authors were to ignore flight, the evolutionary angle is meaningless; if they were to focus on the evolutionary angle (which is lovely), the functionality of the undead neurons must be assessed in the light of flight, not walking. It is possible that the Lineage 0B modulates both flight and walking – locomotion in general. In this case the authors could discuss upfront that walking is being used in their functional imaging experiments because it's an easier prep that will nevertheless report on connectivity of the undead neurons.

Rearranging the manuscript and explaining that octopaminergic neurons are involved in both walking and flight hopefully helps to clear this confusion.

5) Do the authors have data looking at markers of PCD in the bee and swift louseflies? If so please include them. If not, given the challenges of COVID-19 and the difficulty of obtaining these animals we do not request these as new experiments.

We use 2 criteria for detecting dying cells: 1) antibody for active effector caspase Dcp-1 which and 2) the morphology of Dcp-1 labelled cells – they are clearly pyknotic cell remnants indicative of a cell dying and being broken down into bits, see Figure 7I compared with Figure 2—figure supplement 1. We don’t have any other data.